# SUB-TASK DECOMPOSITION ENABLES LEARNING IN SEQUENCE TO SEQUENCE TASKS

**Noam Wies, Yoav Levine & Amnon Shashua**
The Hebrew University of Jerusalem
`{noam.wies,yoav.levine,shashua}@cs.huji.ac.il`

## ABSTRACT

The field of Natural Language Processing (NLP) has experienced a dramatic leap in capabilities with the recent introduction of huge Language Models (LMs). Despite this success, natural language problems that involve several compounded steps are still practically unlearnable, even by the largest LMs. This complies with experimental failures for end-to-end learning of composite problems that were demonstrated in a variety of domains. An effective mitigation is to introduce intermediate supervision for solving sub-tasks of the compounded problem. Recently, several works have demonstrated high gains by taking a straightforward approach for incorporating intermediate supervision in compounded natural language problems: the sequence-to-sequence LM is fed with an augmented input, in which the decomposed tasks' labels are simply concatenated to the original input (see figure 1). In this paper, we prove a positive learning result that motivates these recent efforts. We show that when concatenating intermediate supervision to the input and training a sequence-to-sequence model on this modified input, unlearnable composite problems can become learnable. We show that this is true for any family of tasks which on the one hand, are unlearnable, and on the other hand, can be decomposed into a polynomial number of simple sub-tasks, each of which depends only on $O(1)$ previous sub-task results. Beyond motivating contemporary empirical efforts for incorporating intermediate supervision in sequence-to-sequence language models, our positive theoretical result is the first of its kind in the landscape of results on the benefits of intermediate supervision for neural-network learning: Until now, all theoretical results on the subject are *negative*, *i.e.*, show cases where learning is impossible without intermediate supervision, while our result is *positive*, showing that learning is facilitated in the presence of intermediate supervision.

## 1 INTRODUCTION

Large-scale language models such as BERT (Devlin et al., 2019), T5 (Raffel et al., 2020), and GPT-3 (Brown et al., 2020) have recently pushed the envelope in many NLP tasks. Nevertheless, there are some problem-families that even the largest models do not seem to be capable of solving. One such family is that of "multi-hop" reasoning problems (see, e.g., Geva et al. (2021); Kalyan et al. (2021); Press et al. (2022)) that require compounding operations in order to produce an answer. For example, Gopher (Rae et al., 2021), one of the largest available language models, achieves 61% accuracy in the StrategyQA benchmark (Geva et al., 2021) that requires implicit decomposition into reasoning steps, while human level performance is around 87% accuracy.

The limitations of learning compounded tasks with neural networks in an end-to-end manner have been observed in a variety of non-linguistic domains. A leading experimental approach for addressing these is to first explicitly break the compounded operations into more basic "single-hop" operations and then combine the results. Gülçehre & Bengio (2016), one of the earliest works on this subject, propose that supervision for the single-hop intermediate steps is crucial for avoiding bad local minima in the optimization of neural networks. Afterward, Glasmachers (2017) demonstrated that gradient-based end-to-end multi-hop learning is inefficient for solving complex problems that are easily solved by a divide-and-conquer strategy. Beyond position papers, specific examples were

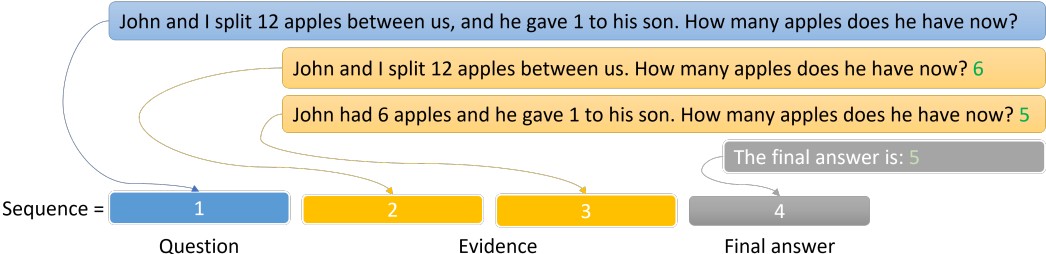

Figure 1: An illustrative example of the prominent method for introducing sub-task decomposition and intermediate supervision for math word problems (Ling et al., 2017; Cobbe et al., 2021). Intermediate sub-tasks and their labels are concatenated to the original task's input to form a new input sequence. At training time, the likelihood of the entire sequence following the original input is maximized conditioned on the input, and at test time only the original input is given to the model.

shown, *e.g.*, Chang et al. (2020) showed that SATNet (Wang et al., 2019) could not solve visual Sudoku without using intermediate labels to identify individual Sudoku digit images.

Similar limitations were observed in language related compounded tasks, including commonsense reasoning (Liu et al., 2022; Wei et al., 2022; Zelikman et al., 2022), math word problems (Piękos et al., 2021; Wei et al., 2022), and programs execution (Nye et al., 2022). The go-to architectures in this domain are powerful language models, which are trained as sequence-to-sequence models over text. In this setting, a particular form of introducing intermediate supervision for compounded tasks has emerged: intermediate sub-tasks and their labels are concatenated to the original task's input to form a new input sequence, on which the sequence-to-sequence LM is trained. This approach has recently been widely adopted, *e.g.*, by Rajani et al. (2019); Cobbe et al. (2021); Piękos et al. (2021); Recchia (2021); Nye et al. (2022); Wei et al. (2022); Zelikman et al. (2022). Figure 1 illustrates this approach for math problems, as done in Ling et al. (2017); Cobbe et al. (2021). These works show that training sequence-to-sequence models with concatenated sub-task decomposition supervision significantly improves the results when compared to training the same model without the intermediate supervision. For example, Nye et al. (2022) show $> 99\%$ accuracy for 8 digits addition when concatenating intermediate calculations to the input, while the vanilla accuracy without intermediate supervision is around $\sim 35\%$.

While such decomposition based approaches are intuitive, we are not aware of theoretical results that motivate and formulate their benefits for learning composite problems with neural-networks. In this paper, we provide positive theoretical results in this domain, which are in fact the first of their kind (see related work in section 2). We show our results for sequential models, integrating the intermediate supervision in a manner that mimics the above cited successful empirical approaches in the language domain. In this formulation, a learner learns to predict a sequence composed of the task inputs $\mathbf{x}$, followed by the single-hop reasoning steps referred to as the *evidence*, and finally, the final answer $y$. We extend provable guarantees for the convergence of overparameterized recurrent neural networks (Wang et al., 2021) and prove that with intermediate sub-task supervision, even a simple sequence-to-sequence model provably learns any task that obeys an efficient decomposition into simpler subtasks that depend only on a small fraction of the input. Importantly, both the sample complexity and the required number of gradient updates are polynomial. In contrast, we rely on existing works (Valiant, 1984; Goldreich et al., 1986; Daniely & Shalev-Shwartz, 2016) to show that in the absence of intermediate supervision, there exist efficiently decomposable tasks that are unlearnable with polynomial time learning algorithms.

Our results apply to a broad family of tasks. As a first exemplifying step, we show a positive result for learning bit subset parity, a setting that is notoriously not amenable to gradient-based algorithms in an efficient way without intermediate supervision (Kearns, 1998; Shalev-Shwartz et al., 2017; Abbe & Sandon, 2020; Abbe et al., 2021). In this setting, the family of target functions consists of parities over subsets of unknown input bits. Specifically, the input is $d$ bits and the task is to predict whether the number of 1's in certain unknown subset of $d/2$ bits is odd or even. The corresponding sub-tasks we consider are the parities of subsets of the unknown input subset. We prove a theorem guaranteeing that, when intermediate supervision is available, efficient neural network learning is

made possible. As a result, we show an exponential gap between the end-to-end and decomposition-based neural network learnability of the bit subset parity problem.

Next, we generalize the above result, and show that when sufficient intermediate supervision is available, any family of functions with a polynomial time complexity, *i.e.*, functions that belong to the **P** time complexity class, are efficiently learnable by neural networks. Accordingly, based on either standard cryptographic assumptions (Valiant, 1984; Goldreich et al., 1986) or computational complexity hardness assumptions (Daniely & Shalev-Shwartz, 2016) we prove that there exist tasks that, on the one hand, cannot be learned by any polynomial time algorithm and, on the other hand, can be efficiently learned by neural networks when intermediate supervision is present.

Our main result can be stated as follows:

**Theorem 1.** *(Informal) There exists a binary classification problem parameterized by size $d$, such that the following holds:*

- *On one hand, when equipped with sub-task decomposition supervision, a simple sequence-to-sequence model can get arbitrarily low $\epsilon > 0$ zero-one loss with number of gradient updates that is **polynomial** in $d, \epsilon^{-1}$.*

- *On the other hand, when supervision regarding sub-task is missing, then for **any** polynomial time learning algorithm and (constant) $\epsilon > 0$, the zero-one loss will be **higher** than $1/2 - \epsilon$.*

To summarize, the main contributions of this paper are:

1. We show the first *positive result* that guarantees neural networks learnability in the presence of intermediate supervision for a problem that is unlearnable without it.

2. We do so in the sequence-to-sequence setting that is currently used for applying state-of-the-art language models on complex multi-hop tasks in NLP.

3. We show that with sufficient intermediate supervision this sequence-to-sequence setting allows learning any function in the **P** time complexity class.

The remainder of this paper is organized as follows. Section 3 presents the sequence to sequence model we analyzed. Section 4 presents a hypothesis class and proves that it can be learned with our sequence to sequence model. Section 5 presents a concrete example, and demonstrates that the task of learning bit-subset parity with sequence-to-sequence models can be learned with sub-task decomposition and the corresponding intermediate supervision. Finally, in section 6 we generalize the positive results to any function in the **P** time complexity class, thus establishing our main result.

## 2 RELATED WORK

The concept of how learning can be enhanced by guiding a learner through intermediate tasks is an old one, dating back to animal training by shaping (Skinner, 1958; Peterson, 2004; Krueger & Dayan, 2009). Since then, a large body of work has shown its practical benefits for various machine learning tasks. For example, there exists a rich line of work on the importance of shaping rewards and adding sub-goals in reinforcement learning tasks. Karlsson (1994) introduced the methodology of using knowledge in the reward function, in order to decompose a holistic task into several sub-tasks. Ng et al. (1999) established necessary and sufficient conditions for reward shaping to reserved optimal policies. Marthi (2007) investigate the problem of automatically learning a decomposition of the reward function. All these work intuitively rely on benefits of adding intermediate supervision. Recently, Zhai et al. (2022) showed that adding sub-goal rewards provably reduces the complexity of the synchronous value iteration algorithm. However, this reduction is linear in the number of the sub-goals, unlike our work that proves exponential gap in the supervised learning setting. Moreover, several of the notions in their analysis are unique to the reinforcement leaning setup and cannot be easily translated into the supervised learning setting (*e.g.*, One-Way Intermediate States).

Negative theoretical results exist, showing that end-to-end learning of multi-hop problems is unfeasible without decomposition. Shalev-Shwartz et al. (2017) explored the theoretical limitations of end-to-end gradients based learning, studying learnability of tasks that are composed of classification and parity tasks, proving that the end-to-end approach does not converge in a polynomial

number of gradient updates. They do show that when intermediate supervision is provided, the gradients have a much higher signal-to-noise ratio. However, they provide no guarantees that in this case learning is possible in a polynomial number of gradient updates. In addition, Shalev-Shwartz & Shashua (2016) proved an exponential gap between end-to-end-based verification sample complexity and the decomposition-based verification sample complexity. However, again, an explicit setting in which providing intermediate supervision for training actually improves the situation to a point that learning is feasible, is lacking. We provide the first theoretical result proving that neural networks also benefit from sub-task decomposition, while earlier theoretical works in this space only prove that end-to-end learning is unfeasible in some compounded cases.

## 3 THE ANALYZED SEQUENCE-TO-SEQUENCE LEARNING ALGORITHM

A recent successful empirical approach for solving compounded natural language problems (Ling et al., 2017; Rajani et al., 2019; Piękos et al., 2021; Recchia, 2021; Cobbe et al., 2021; Nye et al., 2022; Wei et al., 2022; Zelikman et al., 2022) is to concatenate intermediate supervision labels to the input. This way, the language model receives a sequence composed of the input followed by the labels of the intermediate tasks, before emitting the final compounded answer. For a compounded binary classification task which consists of a $d$-bit input string $\mathbf{x}$, with $\mathcal{S}$ denoting the string of intermediate step results, we denote the combined input sequence as $\mathbf{z} = \mathrm{Concat}\{\mathbf{x}; \mathcal{S}\}$, and the combined output sequence as $\mathbf{y}$, defined in a standard autoregressive fashion by[1] $y_t = z_{t+1}$ (see figure 2 for a $d = 8$ example). Training and testing follow conventional sequence-to-sequence model protocol: At training time, $z_t$ for $t > d$ will be the ground-truth sub-task result $y_{t-1}$ (a practice sometimes referred to as "teacher forcing" (Williams & Zipser, 1989)), and at test time, $z_t$ for $t > d$ will be the model's prediction at time $t - 1$.

We analyze the classical Elman recurrent neural networks (Elman, 1990) with ReLU activations as our sequence-to-sequence model. Given an input sequence $\mathbf{z}$ of length $T = d + \mathrm{len}\,(\mathcal{S})$ as defined above, the architecture $f^{\mathrm{RNN}}$ computes:

$$\forall t \in [T] \quad h_t\,(\mathbf{z}) = \mathrm{ReLU}\,(Wh_{t-1} + A\mathbf{e}_{z_t}) \tag{1}$$

$$\forall t \in [T] \quad f_t^{\mathrm{RNN}}\,(\mathbf{z}) = B^T h_t\,(\mathbf{z}) \tag{2}$$

$$h_0\,(\mathbf{z}) = \mathrm{ReLU}\,(M_0) \tag{3}$$

where $\mathbf{e}_0, \mathbf{e}_1 \in \mathbb{R}^2$ are one-hot vectors, $A \in \mathbb{R}^{m \times 2}$ translates the input to the hidden dimension $m$, $W \in \mathbb{R}^{m \times m}$ is the learned hidden weights matrix, $B \in \mathbb{R}^m$ is the output weights vector and $M_0 \in \mathbb{R}^m$ is the initialization of the hidden state.

We will use the binary cross-entropy loss over output locations for $t \geq d$, *i.e.*, our loss ignores the architecture's prediction of $\mathbf{x}$ and depends on its prediction of intermediate labels and final outcome:

$$l\,(\mathbf{y}, \mathbf{s}) = \left(\frac{1}{T - d}\right) \sum_{t=d}^{T} \log\,\left(1 + e^{-y_t \cdot s_t}\right) \tag{4}$$

Algorithm 1 below describes the analyzed training procedure of our sequence-to-sequence model. This algorithm describes a straightforward SGD training procedure where, for simplicity, we analyze a variant that updates only the hidden $W$ weights while keeping $A, B, M_0$ frozen at initialization. This amounts to keeping the input, output and the $t = 0$ hidden state untrained, and training only the core recurrence operation to perform the task while complying with these frozen components.

## 4 COMPOUNDED SEQUENCE TO SEQUENCE LEARNABILITY

In this section, we present a hypothesis class and prove for it that the above described "teacher forcing" (Williams & Zipser, 1989) of intermediate supervision at training time with algorithm 1 provably leads to generalization in polynomial sample complexity and gradient updates. This guarantee will allow us to prove our positive results in the following sections, as we will show that interesting function families belong to this hypothesis class.

---

[1]For clarity, we wrote $y_t = z_{t+1}$ although the inputs domain in our model was $\{0, 1\}$ while the output domain was $\{-1, 1\}$ and hence $y_t = 2 \cdot (z_{t+1} - 1/2)$

---

**Algorithm 1:** Training $f^{\text{RNN}}$ with SGD

---

**Data:** Data set $\mathcal{D}$, learning rate $\eta$.
**Initialization:** The entries of $W^{(0)}, A, M_0$ are i.i.d. generated from $N(0, \frac{2}{m})$. The entries of $B$
  are i.i.d. generated from $N(0, \frac{1}{m})$.
**for** $i = 1, 2, 3...n$ **do**
  | Randomly sample $(\mathbf{x}_i, \mathbf{y}_i)$ from the data set $\mathcal{D}$.
  | $W^{(i)} = W^{(i-1)} - \eta \nabla_{W^{(i-1)}} \ell(\mathbf{y}_i, f^{\text{RNN},W^{(i-1)}}(\mathbf{z}_i))$.
**end**

---

In order to analyze the teacher forcing technique, we begin with an important observation. Essentially, we show that when the zero-one loss of all the single-hop sub-tasks is low, then it implies that also at test time, when the model does not have the ground truth results of the previous sub-tasks and the errors might accumulate, the zero-one loss on the final answer is still low:

**Lemma 1.** *Denote by $z_t^{train} := y_{t-1}$ the ground truth input at training time, and by $z_t^{test} := f_{t-1}^{RNN,W}(\mathbf{z}^{test})$ the iteratively predicted input at test time. Then, for any $W$ the following holds:*

$$\mathbb{E}_{\mathbf{x}}\left[l_{0-1}\left(y, f_T^{RNN,W}\left(\mathbf{z}^{test}\right)\right)\right] \leq \mathbb{E}_{\mathbf{x}}\left[\sum_{t=d}^{T} l_{0-1}\left(y_t, f_t^{RNN,W}\left(\mathbf{z}^{train}\right)\right)\right] \tag{5}$$

*Proof.* Clearly, for any $\mathbf{x}$ when $f^{\text{RNN},W}\left(\mathbf{z}^{\text{train}}\right)$ solves all of the sub-tasks correctly we have that $\mathbf{z}^{\text{test}} = \mathbf{z}^{\text{train}}$ and therefore they have the same zero zero-one loss. So it is enough to upper bound the probability that $f^{\text{RNN},W}\left(\mathbf{z}^{\text{train}}\right)$ is erroneous in any sub-task. Now by definition, for any $t$ the zero-one loss at the $t$'th location is equal to the probability of wrong prediction at this location. Therefore, by the union bound, we get that the sum of the zero-one loss over all the locations is upper bounding the probability of $f^{\text{RNN},W}\left(\mathbf{z}^{\text{train}}\right)$ make an error in any sub-task. See full details in section A at the appendix. $\qquad\square$

As expected, due to a union bound, when the model does not have the ground truth results of the previous sub-task the error can increase by a factor of $T - d$ but this increase is relatively modest as long as $T$ is polynomial in $d$.

Lemma 1 above assures us that it is enough to find an hypothesis class for which algorithm 1 converges and generalizes when we do have the ground truth results of the previous sub-tasks, in order to prove that the teacher forcing technique works. As a candidate for such a hypothesis class, we consider tasks for which the output at each location $d \leq t \leq T$ can be written as sign of composition of linear functions (represented by $\mathbf{w}$ below) of at most $N < T$ input locations $j_1, \ldots, j_N \leq t$, with polynomials activations $\psi_t(x) = \sum_{i=0}^{\deg(\psi_t)} a_{t,i} x^i$:

$$\forall d \leq t \leq T \quad h_t(\mathbf{z}) = \text{sign}\left(\psi_t\left(\left\langle \frac{\mathbf{w}^{(t)}}{\|\mathbf{w}^{(t)}\|}, \begin{pmatrix} \mathbf{e}_{z_{j_1}} \\ \vdots \\ \mathbf{e}_{z_{j_N}} \end{pmatrix} \right\rangle\right)\right) \tag{6}$$

In order to prove convergence and generalization results, we will measure the complexity of functions in the above hypothesis class by a function $\phi(T, \psi, N)$, described formally in appendix A. Importantly, $\phi(T, \psi, N)$ is polynomial in both $T$ and $\max_{t,i}|a_{t,i}|$, while exponential in both $\max_t \deg(\psi_t)$ and $N$. We will denote by $\mathcal{H}_{\phi(T,\psi,N)}$ the hypothesis class described in eq 6.

Now, with this hypothesis class, we can combine lemma 1 with theorem 2 in Wang et al. (2021). They study the learnability of RNNs for binary classification tasks[2] without intermediate supervision, and prove that algorithm 1 is capable of learning function where the final answer $y$ have low complexity $\phi(T, \psi, N)$.

---

[2] They also discussed the case where $y$ is a sequence, however, in their case it was considered part of the task, unlike our case where we add intermediate steps as a method of supervision.

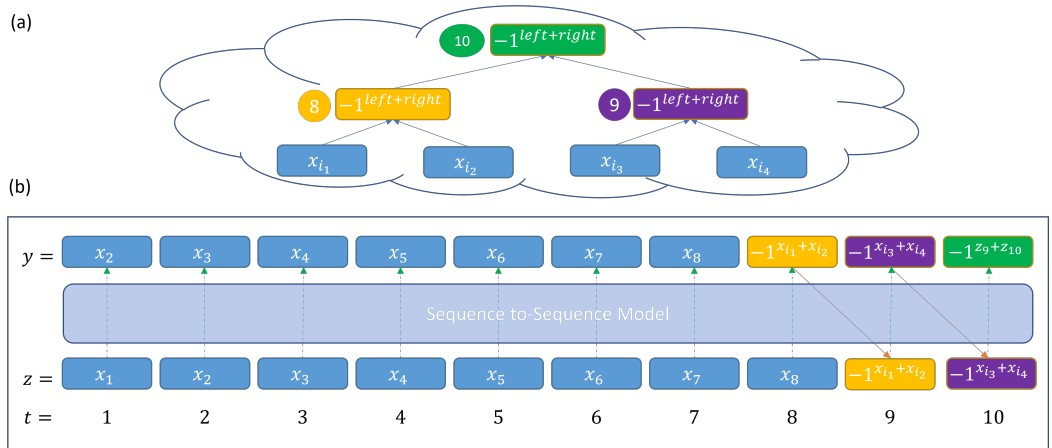

Figure 2: Illustration of the proposed input and output for learning the $d = 8$ bit-subset parity problem with sequence-to-sequence models.

**Theorem 2.** *Denote by $z_t^{test} := f_{t-1}^{RNN,W}\left(\mathbf{z}^{test}\right)$ the iteratively predicted input at test time, and let $\epsilon, \delta > 0$. Assume we run Algorithm 1 for $n > \tilde{O}\left(\left(\frac{\phi(T,\psi,N)+\log\left(\frac{1}{\delta}\right)}{\epsilon}\right)^2\right)$ iterations with learning rate $\eta = \frac{1}{m\sqrt{n}}$, then there exists $m^\star = poly\left(n, \delta^{-1}, T\right)$ such that if $m > m^\star$ then for any $h \in \mathcal{H}_{\phi(T,\psi,N)}$ with probability at least $1 - \delta$ over the randomness in Algorithm 1, the following holds:*

$$\frac{1}{n}\sum_{i=1}^{n}\mathbb{E}_\mathbf{x}\left[l_{0-1}\left(y, f_T^{RNN,W^{(i)}}\left(\mathbf{z}^{test}\right)\right)\right] < \epsilon \tag{7}$$

*where $W^{(i)}$ denotes the output of Algorithm 1 at the $i$'th iteration and $l_{0-1}$ is the zero-one loss.*

Note that sections C,D of the appendix extends theorem 2 for both SGD and GD with finite precision.

In the next sections, we will prove our positive results by showing the intermediate single-hop sub-tasks of the analyzed tasks belong to $\mathcal{H}_{\phi(T,\psi,N)}$ with low complexity $\phi(T,\psi,N)$.

## 5 LEARNING BIT-SUBSET PARITY WITH SEQUENCE TO SEQUENCE MODELS

As a concrete demonstration, in this section we show that unlike the end-to-end case, bit-subset parity *can be learned* with neural networks when intermediate supervision is provided. We begin by defining the challenging task of learning parities over unknown subsets of input bits. Specifically, for a $d$-bit string with a subset of $d/2$ randomly predefined unique indices $i_1, \ldots, i_{d/2}$, our goal is to train a predictor mapping $\mathbf{x} \in \{0,1\}^d$ to $y = (-1)^{\sum_{j=1}^{d/2} x_{i_j}}$ where $\mathbf{x}$ is uniformly distributed. In words, $y$ indicates whether the number of 1's in the given subset of coordinates of $\mathbf{x}$ is odd or even.

We analyze this task as a "multi-hop" task by decomposing it into natural intermediate sub-tasks: parities of subsets of the predefined input subset $x_{i_1}, ..., x_{i_{d/2}}$. Concretely, assuming for simplicity that $d/2$ is a power of 2, and beginning with only two adjacently indexed input bits at a time, we recursively treat the parity of every two adjacently indexed subgroups as an intermediate task. Figure 2(a) illustrates this binary tree sub-task decomposition of our parity problem. The leaves of the tree of intermediate labels $\mathcal{T}$ are $-1^{x_{i_1}+x_{i_2}}, \ldots, -1^{x_{i_{d/2-1}}+x_{i_{d/2}}}$ and each node in the tree represents the sub-task of calculating the parity function over its descendants.

In order to fit into the sequence-to-sequence setting of section 3, we translate our imaginary tree of intermediate labels $\mathcal{T}$ into a sequence of intermediate labels $\mathcal{S}$ by inverse-BFS like tree traversal, and then concatenate the sequence $\mathcal{S}$ after the input $\mathbf{x}$. An exact formulation of the mapping from tree $\mathcal{T}$ to sequence of intermediate labels $\mathcal{S}$ is given in appendix B.

### 5.1 Learnability of Bit-Subset Parity With and Without Intermediate Supervision

In this section we show that the sub-tasks of learning bit-subset parity are simple enough to be covered by the result in theorem 2, *i.e.*, we prove that our sequence-to-sequence formulation of the bit-subset parity target function, which includes the intermediate labels sequence $\mathcal{S}$ defined above, can be written as a multivariate function where each of its outputs is a simple low degree polynomial of at most $O(1)$ inputs bits. We show that this is indeed the case, *i.e.*, that all the parity target functions comprising the intermediate supervision to our problem belong to $\mathcal{H}_{\phi(T,\psi,N)}$ (see section 4) with $N$, $\max_t \deg(\psi_t)$, $\max_{t,i} |a_{t,i}|$ that do not grow with $d$. Importantly, when defining the input sequence to be only the original input, without the concatenated sub-task decomposition labels, then the function $h_T(\mathbf{x})$ clearly depends on $d/2$ bits, and therefore will require $N = d/2$, that leads to exponential complexity $\phi(T,\psi,N) = \Omega(e^d)$. Thus, no efficient learning is guaranteed for the original compounded task.

We begin by showing that our single-hop tasks of parities over two bits (see illustration in figure 2(a)) are simple degree-2 polynomials:

**Lemma 2.** *There exists degree two polynomial $\psi(x) = a_2 x^2 + a_1 x + a_0$ with bounded coefficients $\forall i \quad |a_i| < 10$ as well as $\mathbf{w} \in \mathbb{R}^4$ such that:*

$$\forall z_1, z_2 \in \{0,1\} \quad \psi\left(\left\langle \frac{\mathbf{w}}{\|\mathbf{w}\|}, \begin{pmatrix} \mathbf{e}_{z_1} \\ \mathbf{e}_{z_2} \end{pmatrix} \right\rangle\right) = \begin{cases} 1 & z_1 = z_2 \\ -1 & z_1 \neq z_2 \end{cases} \tag{8}$$

*Proof.* We will use $\mathbf{w}$ to sum the first coordinates of $\mathbf{e}_{z_1}, \mathbf{e}_{z_2}$, and use polynomial interpolation to find a degree two polynomial $\psi$ that interpolates the $z_1 = z_2 = 0$, $z_1 \neq z_2$, $z_1 = z_2 = 1$ points, see full details in appendix B. $\qquad\square$

The above lemma implies that all of the target functions in our defined intermediate supervision belong to $\mathcal{H}_{\phi(T,\psi,N)}$ for $\phi(T,\psi,N) = O(d)$. Therefore, together with theorem 2, it assures us that when intermediate supervision is available, Algorithm 1 can learn bit-subset parities with polynomial network size, sample complexity and number of gradient updates.

Now, after we showed that when incorporating intermediate supervision bit-subset parities can be learned by a neural network, we will use the results of Shalev-Shwartz et al. (2017) to establish an exponential gap between the end-to-end and decomposition-based neural network learnability[3]:

**Corollary 1.** *When learning bit-subset parities using neural networks, the following holds:*

- *On one hand, when equipped with sub-task decomposition supervision, a simple sequence-to-sequence model can get arbitrarily low $\epsilon > 0$ zero-one loss with number of gradient updates that is **polynomial** in $d, \epsilon^{-1}$.*

- *On the other hand, when supervision regarding sub-task is missing, then for any (constant) $\epsilon > 0$ with high probability over the target parity, the zero-one loss will be **higher** than $1/2 - \epsilon$ unless the number of gradient updates is **exponential** in $d$.*

*Proof.* Follows directly by combining theorem 2 and lemma 2 with the the negative results in Shalev-Shwartz et al. (2017). See full details in section F at the appendix. $\qquad\square$

### 5.2 Bit-Subset Parity Experiments

In section 5.1 we proved an exponential gap when using Elman RNNs (Elman, 1990) to learn bit-subset parity with and without sub-task decomposition. This section empirically demonstrates that the same gap exists with the commonly used Transformer (Vaswani et al., 2017) architecture. We trained a series of models while varying the input sizes from 8 bits to 256 bits. For each input size,

---

[3] Note that the negative result holds only for full gradient descent and does not hold for stochastic gradient descent, for which Abbe & Sandon (2020); Abbe et al. (2021) show that parities are efficiently learnable when using complex non-random initialization.

we trained a BERT-base sized Transformer model for $100k$ iterations[4] with and without intermediate supervision. The intermediate supervision was introduced exactly as described in the previous subsection, see Figure 2 for an illustration. See full technical details of the training apparatus in appendix G.

Figure 3 clearly shows that in a practical setting, using common Transformer networks, a very large gap quickly opens between the settings with and without intermediate supervision. The employed BERT base sized Transformer architecture is a strong network that pushed the envelope on very challenging NLP tasks, and is much stronger than the theoretically analyzed RNN. Still, learning even the 32 bit subset parity task without supervision proved to be too challenging even for this network (no learning after over 2M steps), while it easily learned the task in the presence of intermediate supervision. Overall this experiment, performed on the same task on which we prove our theoretical results, reinforces their relevance to common Transformer architectures.

## 6 UNIVERSALITY OF DECOMPOSITION BASED SEQUENCE-TO-SEQUENCE LEARNING

In this section, we prove our main result (outlined in the introductory theorem 1). On the one hand, we generalize the positive results of section 5.1 by showing that when sufficient intermediate supervision is available, a neural network can efficiently learn any function in the $\mathbf{P}$ time complexity class. On the other hand, we rely on existing works (Valiant, 1984; Goldreich et al., 1986; Daniely & Shalev-Shwartz, 2016) to show that under either standard cryptographic assumptions or computational complexity hardness assumptions, there exist functions in the $\mathbf{P}$ time complexity class that cannot be learned by any polynomial time learning algorithm without intermediate supervision.

We begin by defining the decomposition of any function $f$ in the $\mathbf{P}$ time complexity class into sub tasks. For that, we will use the fact that any such $f$ has polynomial circuit complexity (see for example theorem 9.30 in Sipser (2013)), and therefore can be computed by a boolean circuit with polynomial size. We will denote by $G = (V, E)$ the directed acyclic graph associated with such a circuit, and by $l_v$ the logic gates of each vertex $v$. Furthermore, since both the "AND" and "OR" logical gates can be decomposed into a boolean circuit with binary-tree like structure, we may assume that the input degree of each vertex is $O(1)$.

Now, in order to fit into the sequence-to-sequence setting of section 3, we define the intermediate labels sequence $\mathcal{S}$ for any $f$. Basically, each non-leaf vertex $v \in V$ will represent an intermediate task with its ground-truth label determined by $l_v$, and we will use a topological sorting of $G$ in order to translate $G$ into a sequence of intermediate labels $\mathcal{S}$ with length $T := |V|$ (see figure 2 for a concrete example of this abstract construction strategy). Importantly, as in the bit-subsets parity task, $T$ is polynomial in $d$.

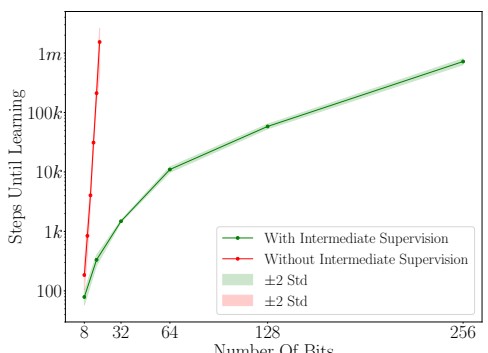

In order to show our generalized positive result, theorem 2 motivates us to prove that our sequence-to-sequence formulation of any function $f$ in the $\mathbf{P}$ time complexity class, which includes the intermediate labels sequence $\mathcal{S}$ defined above, can be written as a multivariate function where each of its outputs is a simple low degree polynomial of at most $O(1)$ input bits. Lemma 3 below shows that this is indeed the case, *i.e.*, that all the target functions comprising the intermediate supervision to our

Figure 3: The number of steps until a BERT-base sized Transformer learns bit-subset parities with and without intermediate supervision. By learning we mean validation accuracy higher than 60%. While this definition is somehow arbitrary, in practice we observed a grokking phenomenon (Power et al., 2021) where very soon after the accuracy became higher than random level it also became almost perfect (accuracy $> 95\%$).

problem belong to $\mathcal{H}_{\phi(T,\psi,N)}$ (see section 3) with $N$, $\max_t \deg(\psi_t)$, $\max_{t,i} |a_{t,i}|$ that do not grow with $d$.

---

[4] With the exception of the 16 and 18 bits tasks without intermediate supervision, for which we increase the number of iterations to 300K and 2M respectably in order to try and successfully learn the task.

**Lemma 3.** *For any logical gate $l_v : \{0,1\}^N \to \{0,1\}$ with $N = O(1)$, there exists $O(1)$ degree polynomial $\psi(x) = \sum_{i=0}^{deg(\psi)} a_i x^i$ with bounded coefficients $\max_i |a_i| = O(1)$ as well as $\mathbf{w} \in \mathbb{R}^{2N}$ such that:*

$$\forall z_1, \dots, z_N \in \{0,1\} \quad \psi\left(\left\langle \frac{\mathbf{w}}{\|\mathbf{w}\|}, \begin{pmatrix} \mathbf{e}_{z_1} \\ \vdots \\ \mathbf{e}_{z_N} \end{pmatrix} \right\rangle\right) = l_v(z_1, \dots, z_N) \tag{9}$$

*Proof.* We will use $\mathbf{w}$ to uniquely represent each possible combination of $z_0, \dots, z_N$ as an $N$ bit real value number, and use polynomial interpolation to find a $2^N$-degree polynomial $\psi$ that interpolates the $z_1 = \cdots = z_N = 0, \dots, z_1 = \cdots = z_N = 1$ points. Finally the $O(1)$ coefficients boundedness follow from taking maximum[5] over all possible $l_v$ logical gates. see full details in appendix B. $\square$

The above lemma implies that all of the target functions in our defined intermediate supervision belong to $\mathcal{H}_{\phi(T,\psi,N)}$ for $\phi(T, \psi, N) = O(d)$. Therefore, together with theorem 2, it assures us that when intermediate supervision is available, Algorithm 1 can learn any function in the **P** time complexity class with polynomial network size, sample complexity and number of gradient updates.

Now, after we showed that when incorporating intermediate supervision any function in the **P** time complexity class can be learned by a neural network, our main results is a simple corollary of the above results:

**Corollary 2.** *Under either standard cryptographic assumptions or computational complexity hardness assumptions, there exists a binary classification problem parameterized by size $d$, such that the following holds:*

- *On one hand, when equipped with sub-task decomposition supervision, a simple sequence-to-sequence model can get arbitrarily low $\epsilon > 0$ zero-one loss with number of gradient updates that is **polynomial** in $d, \epsilon^{-1}$.*

- *On the other hand, when supervision regarding sub-task is missing, then for **any** polynomial time learning algorithm and (constant) $\epsilon > 0$, the zero-one loss will be **higher** than $1/2 - \epsilon$.*

*Proof.* Follows directly by combining theorem 2 and lemma 3 with either the negative results in Valiant (1984); Goldreich et al. (1986) or in Daniely & Shalev-Shwartz (2016). $\square$

## 7 DISCUSSION

In this paper, we show for a broad family of functions an exponential gap between learning algorithms that rely on intermediate supervision and algorithms that do not rely on intermediate supervision. Across domains and architectures, there has been a wide range of proposed methods for introducing intermediate supervision. Some design specialized architectures, some add relevant loss terms, etc. The method that is taking over in the NLP domain is straightforward, and is particularly natural for this domain in which the core architectures are strong sequence-to-sequence Language Models: Concatenate the intermediate supervision to the input, and thus jointly train the model to maximize the likelihood of all the intermediate labels as well as the overall output. Our analysis is framed in this space, and motivates this intuitive incorporation of intermediate supervision in the framework of sequence-to-sequence models. We show that even with a simple sequence-to-sequence architecture it is feasible to expect such simultaneous compounded learning to be useful. In this regard, we view our work as providing timely theoretical feedback to the rapid empirical advances in this field.

**Limitations:** We proved universal learnability results when sufficient intermediate supervision was provided. A fundamental question is what happens when we limit the amount of sub-task supervision. For the task of bit-subset parity, we demonstrated that supervision regarding $O(d)$ sub-tasks can yield an exponential advantage. An interesting question that we leave open for future work is whether there exists a similar advantage with only $O(1)$ sub-tasks.

In addition, while our results show an exponential gain, it is still unclear which sub-tasks are solvable by end-to-end methods, and which tasks require decomposition? Interestingly, a recent study (Abbe et al., 2022) addressed exactly this question for one-layer hidden networks in the mean-field regime. However, our understanding of this question for practical architectures is still very limited.

---

[5]This maximum exists since there is a finite number of possible functions from $\{0,1\}^N$ into $\{0,1\}$.

## REPRODUCIBILITY STATEMENT

A complete proof of all the theoretical claims was included in the appendix. We also provide the source code for the bit-subset parity experiment in https://github.com/HUJI-Deep/sub_task_decomposition.

## ACKNOWLEDGMENTS AND DISCLOSURE OF FUNDING

We thank Eran Malach and Shai Shalev-Shwartz for a helpful discussion on our stronger negative results, as well as Lifu Wang for clarifying Wang et al. (2021). This research was supported by the ERC (European Research Council) and the ISF (Israel Science Foundation). Yoav Levine was supported by the Israel Academy of Sciences Adams fellowship.

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

## A  COMPOUNDED SEQUENCE TO SEQUENCE LEARNABILITY DETAILS

We start by formally defining our sequence-to-sequence functions complexity measure as:

$$\phi\left(T,\psi,N\right) \coloneqq \tilde{O}\left(T^{16+3N+\max_t \deg(\psi_t)} C^{2N} \max_t \deg\left(\psi_t\right)^{3N} \max_{t,i}\left|a_{t,i}\right|^2\right) \tag{10}$$

Where $C > 0$ is some constant.

Now we prove lemma 1 from the main text. Essentially this lemma applies the union bound to show that when the zero-one loss of all the single-hop sub-tasks is low, then also at test time — when the model does not have the ground truth results of the previous sub-task and errors may accumulate — the zero-one loss on the final answer is still low.

*Proof.* Denote by $\epsilon$ the zero-one loss for $\mathbf{z}^{\text{train}}$, *i.e.*, the right hand side in eq 5. Clearly, for any $\mathbf{x}$ when $f^{\text{RNN},W}\left(\mathbf{z}^{\text{train}}\right)$ solves all the sub-tasks correctly we have that $\mathbf{z}^{\text{test}} = \mathbf{z}^{\text{train}}$ and therefore $l_{0-1}\left(y, f_T^{\text{RNN},W}\left(\mathbf{z}^{\text{test}}\right)\right) = 0$. So it is enough to upper bound the probability that $f^{\text{RNN},W}\left(\mathbf{z}^{\text{train}}\right)$ makes an error in any sub-task by $\epsilon$. But by the zero-one loss definition, for any $t$ we have that:

$$P_{\mathbf{x}}\left(f_t^{\text{RNN},W}\left(\mathbf{z}^{\text{train}}\right) \neq y_t\right) = \mathbb{E}_{\mathbf{x}}\left[l_{0-1}\left(y_t, f_t^{\text{RNN},W}\left(\mathbf{z}^{\text{train}}\right)\right)\right] \tag{11}$$

And therefore $\sum_{t=d}^T P_{\mathbf{x}}\left(f_t^{\text{RNN},W}\left(\mathbf{z}^{\text{train}}\right) \neq y_t\right) = \epsilon$. Finally, by the union bound we got that

$$P_{\mathbf{x}}\left(\exists d \leq t \leq T \quad f_t^{\text{RNN},W}\left(\mathbf{z}^{\text{train}}\right) \neq y_t\right) \leq \sum_{t=d}^T P_{\mathbf{x}}\left(f_t^{\text{RNN},W}\left(\mathbf{z}^{\text{train}}\right) \neq y_t\right) = \epsilon \tag{12}$$

$\square$

## B  SUB TASKS LEARNABILITY PROOFS

In this section we prove lemmas 2,3 from the main text, *i.e.*, we prove that our intermediate steps are simple enough to be covered by theorem 2.

We begin by formally describing the details of learning parities with sequence-to-sequence models. Since sequence-to-sequence models expect their inputs to be sequences, we will translate the tree described in section 5 into a sequence by inverse BFS like tree traversal, and concatenate the result sequence after $\mathbf{x}$. Therefore, our inputs sequence includes all the $d$ variables in $\mathbf{x}$, together with all the sub-tasks decomposition nodes in the binary tree except the root (that represent $y$). So we will have an input sequence length $T$ that is equal to:

$$T \coloneqq d + \text{nodes in full binary tree with } \frac{d}{4} \text{ leaves} - 1 = \frac{3}{2}d - 2 \tag{13}$$

And the ground-truth sub-task results are recursively defined by:

$$\forall t \geq d \quad y_t = \begin{cases} (-1)^{x_{i_2(t-d)+1} + x_{i_2(t-d)+2}} & t < \frac{5}{4}d \\ (-1)^{y_2\left(t-\frac{5d}{4}\right)+d} + y_2\left(t-\frac{5d}{4}\right)+d+1 & \text{else} \end{cases} \tag{14}$$

For $t > d$, at training time, $z_t$ will be the ground-truth sub-task result $y_{t-1}$. At test time $z_t$ will be the model prediction at time $t-1$:

$$\forall t \in [T] \quad z_t = \begin{cases} x_t & t \leq d \\ \frac{1}{2} + \frac{1}{2}\text{sign}\left(f_{t-1}^{\text{RNN}}\left(z_1, \cdots, z_{t-1}\right)\right) & t > d \wedge \text{test} \\ \frac{1+y_{t-1}}{2} & t > d \wedge \text{training} \end{cases} \tag{15}$$

Note that $f^{\text{RNN}}$ is causal model, *i.e.* $f_{t_1}^{\text{RNN}}$ does not depend on $\mathbf{z}_{t_2}$ for any $t_2 > t_1$, and therefore eq 15 is well defined.

Now, we prove lemma 2 from the main text. Essentially this lemma shows that the intermediate steps of length 2 parities belong to the hypothesis class define in eq 6.

*Proof.* Define $\mathbf{w} := \frac{1}{\sqrt{2}} \begin{pmatrix} 1 \\ 0 \\ 1 \\ 0 \end{pmatrix}$, then

$$\left\langle \frac{\mathbf{w}}{\|\mathbf{w}\|}, \begin{pmatrix} \mathbf{e}_{z_1} \\ \mathbf{e}_{z_2} \end{pmatrix} \right\rangle = \begin{cases} \sqrt{2} & z_1 = 0 \wedge z_2 = 0 \\ \frac{1}{\sqrt{2}} & z_1 \neq z_2 \\ 0 & z_1 = 1 \wedge z_2 = 1 \end{cases} \tag{16}$$

Therefore, it is enough to find $\psi$ such that

$$\psi(0) = \psi\left(\sqrt{2}\right) = 1 \wedge \psi\left(\frac{1}{\sqrt{2}}\right) = -1 \tag{17}$$

Finally, we will use Lagrange basis functions to find the required polynomial and get:

$$\psi(z) = \left(\frac{z - \frac{1}{\sqrt{2}}}{0 - \frac{1}{\sqrt{2}}}\right)\left(\frac{z - \sqrt{2}}{0 - \sqrt{2}}\right) - \left(\frac{z - 0}{\frac{1}{\sqrt{2}} - 0}\right)\left(\frac{z - \sqrt{2}}{\frac{1}{\sqrt{2}} - \sqrt{2}}\right) + \left(\frac{z - 0}{\sqrt{2} - 0}\right)\left(\frac{z - \frac{1}{\sqrt{2}}}{\sqrt{2} - \frac{1}{\sqrt{2}}}\right) \tag{18}$$

$$= 1 \cdot \left(z - \frac{1}{\sqrt{2}}\right)\left(z - \sqrt{2}\right) + \frac{z}{2}\left(z - \sqrt{2}\right) + 3z\left(z - \frac{1}{\sqrt{2}}\right) \tag{19}$$

$$= \left(z^2 - \frac{3}{\sqrt{2}}z + 1\right) + \left(\frac{z^2}{2} - \frac{1}{\sqrt{2}}z\right) + \left(3z^2 - \frac{3z}{\sqrt{2}}\right) \tag{20}$$

$$= \frac{9}{2}z^2 - \frac{7}{\sqrt{2}}z + 1 \tag{21}$$

$\square$

Now we prove lemma 3 from the main text. Essentially this lemma shows that also our intermediate steps for any functions in the **P** time complexity class belongs to the hypothesis class define in eq 6.

*Proof.* Denote $\alpha(z_1, \ldots, z_N) := \sum_{i=0}^{N-1} 2^i \cdot 1_{z_i=0}$ the function that converts $N$ bits to their binary string, and define $\mathbf{w} := \sqrt{\frac{3}{4^N - 1}} \begin{pmatrix} 2^0 \\ 0 \\ 2^1 \\ 0 \\ \vdots \\ 2^{N-1} \\ 0 \end{pmatrix}$, then $\mathbf{w}$ is a unit vector that represents $z_1, \ldots, z_N$ as $N$

bit numbers $\left\langle \frac{\mathbf{w}}{\|\mathbf{w}\|}, \begin{pmatrix} \mathbf{e}_{z_1} \\ \vdots \\ \mathbf{e}_{z_N} \end{pmatrix} \right\rangle = \alpha(z_1, \ldots, z_N)$. Now, we can use the Lagrange basis functions to find the required polynomial:

$$\psi(x) = \sum_{z_1, \ldots, z_N = 0}^{1} \left( f_v(z_1, \ldots, z_N) \prod_{(\tilde{z}_1, \ldots, \tilde{z}_N) \neq (z_1, \ldots, z_N)} \left( \frac{x - \alpha(\tilde{z}_1, \ldots, \tilde{z}_N)}{\alpha(z_1, \ldots, z_N) - \alpha(\tilde{z}_1, \ldots, \tilde{z}_N)} \right) \right) \tag{22}$$

Finally the $O(1)$ coefficients boundedness follows from taking the maximum[6] over all possible $f_v$ functions. $\square$

---

[6]This maximum is exists since there is a finite number of possible function from $\{0, 1\}^N$ into $\{0, 1\}$.

## C   EXTENSION FOR SGD WITH FINITE PRECISION

In this section, we prove theorem 2 from the main text holds also for algorithm 2 which is a finite-precision variant of SGD[7]. . We will follow the proof in Wang et al. (2021) while taking into account the finite precision gradients.

---

**Algorithm 2:** Training $f^{\text{RNN}}$ with finite precision SGD (an finite precisio variant of algorithm 1)

**Data:** Data set $\mathcal{D}$, learning rate $\eta$, finite precision $\sigma$.
**Initialization:** The entries of $W^{(0)}, A, M_0$ are generated i.i.d. from $N(0, \frac{2}{m})$. The entries of $B$ are generated i.i.d. from $N(0, \frac{1}{m})$.
**for** $i = 1, 2, 3...n$ **do**
  Randomly sample $(\mathbf{x}_i, \mathbf{y}_i)$ from the data set $\mathcal{D}$.
  Get arbitrary $\sigma$-approximation of the gradient:
  $G^{(i)} \in \mathcal{B}_\infty{}^8 \left( \nabla_{W^{(i-1)}} \ell(\mathbf{y}_i\,,\, f^{\text{RNN}, W^{(i-1)}}(\mathbf{z}_i))\,,\, \sigma \right)$.
  Update weights:
  $W^{(i)} = W^{(i-1)} - \eta G^{(i)}$.
**end**

---

We begin by stating theorem 2[9] in Wang et al. (2021) with our notations:

**Theorem 3.** *Let $\delta > 0$, and assume we run algorithm 1 for $n$ iterations with learning rate $\eta = \frac{1}{m\sqrt{n}}$. Then there exists $m^\star = poly\left(n, \delta^{-1}, T\right)$ such that if $m > m^\star$ then for any $h \in \mathcal{H}_{\phi(T,\psi,N)}$ with probability at least $1 - \delta$ over the randomness in algorithm 1, the following hold:*

$$\mathbb{E}_{\mathbf{x}} \left[ \left( \frac{1}{n(T-d)} \right) \sum_{t=d}^{T} \sum_{i=1}^{n} l_{0-1} \left( y_t, f_t^{RNN, W^{(i)}}(\mathbf{z}) \right) \right] < \tilde{O} \left( \frac{\phi(T, \psi, N)}{\sqrt{n}} \right) + O \left( \frac{\log \frac{1}{\delta}}{\sqrt{n}} \right) \quad (23)$$

*where $W^{[i]}$ denote the output of algorithm 1 at the $i$'th iteration and $l_{0-1}$ is the zero-one loss.*

Clearly when using infinite precision SGD theorem 2 follows from theorem 3 and lemma 1 by simple algebraic manipulations. Therefore, for proving theorem 2 with finite precision gradient based optimization, it is enough to modify theorem's 3 proof to analyze algorithm 2 that uses finite precision gradients, instead of algorithm 1 that uses full precision gradients.

While complicated, Wang et al. (2021)'s proof for theorem 3 can be divided into two high-level arguments. The first argument measure the *complexity* of the learned hypothesis class with respect to random initialization of $f^{\text{RNN}}$ (lemma 6). And the second argument is a generalization bound for algorithm 1 with networks that are overparameterized enough. Since the first argument is independent of the gradients, the proof of the first argument still holds and we only need to prove a generalization bound for algorithm 2. More specifically we only need to prove a lemma that is equivalent to lemma 14 in Wang et al. (2021) and the rest of the second argument (lemma 7 in Wang et al. (2021)) remain unchanged.

**Lemma 4.** *Let $n \in \mathbb{N}$, and denote by $L_i(W) := l\left(\mathbf{y}^{(i)}, f^{RNN,W}(\mathbf{z}^{(i)})\right)$ the training loss. Suppose there exists $W^\star \in \mathcal{B}^{10}\left(W^{(0)}, \frac{R}{\sqrt{m}}\right)$ such that $R = O\left(poly\left(T\right)\right) = \Omega\left(T^{16}\right)$ and $L_i(W^\star) \leq \frac{1+R^2}{n}$. Then for any $\delta > 0$ there exists $m^\star = poly\left(n, R, T, \delta^{-1}\right)$ such that if $m > m^\star$ then with probability at least $1 - \delta$ algorithm 2 with $\eta = \frac{1}{m\sqrt{n}}$ and finite precision $\sigma = O\left(\frac{1}{m}\right)$ will output:*

$$\mathbb{E}_{\mathbf{x}} \left[ \left( \frac{1}{n(T-d)} \right) \sum_{t=d}^{T} \sum_{i=1}^{n} l_{0-1} \left( y_t, f_t^{RNN, W^{(i)}}(\mathbf{z}) \right) \right] < O \left( \frac{R^2 + \log \frac{1}{\delta}}{\sqrt{n}} \right) \quad (24)$$

---

[7]See section D for the extension of the proof to algorithm 3 that is based on GD.

[8]The ball is with respect to the distance that defined by the max matrix norm, *i.e.* the elementwise distances are at most $\sigma$.

[9]Combined with Remark H.1 in Wang et al. (2021)'s supplementary materials. In addition, for simplicity, we state a $1/\sqrt{n}$ convergence rate as opposite to the linear convergence rate in Wang et al. (2021).

[10]The ball is with respect to the distance that defined by the Frobenius matrix norm.

*Proof.* We begin by showing that with high probability over the initialization of $f^{\text{RNN}}$, at any iteration $i \leq n$ of algorithm 2, the distance of the learned hidden weights matrix $W^{(i)}$ from its initialization point $W^{(0)}$ is not too large. As a results we will get that the assumption of lemma 8 uphold, and therefore its upper bound of the deviation from linearization is valid.

By the triangle inequality for any $0 \leq i < n$ we have that:

$$\left\| W^{(i+1)} - W^{(0)} \right\|_F \leq \sum_{k=0}^{i} \left\| W^{(k+1)} - W^{(k)} \right\|_F \tag{25}$$

Substituting algorithm 2 update rule for $W^{(k+1)}$, we get that there exist $\|\sigma_i\|_\infty < \sigma$ such that:

$$W^{(k+1)} - W^{(k)} = -\eta \left( \nabla_{W^{(k)}} \ell(\mathbf{y}_k \, , \, f^{\text{RNN},W^{(k)}}(\mathbf{z}_k)) + \sigma_k \right) \tag{26}$$

Now, explicitly writing $\nabla_{W^{(k)}} \ell(\mathbf{y}_k \, , \, f^{\text{RNN},W^{(k)}}(\mathbf{z}_k))$ with the chain rule we have that:

$$\nabla_{W^{(k)}} \ell(\mathbf{y}_k \, , \, f^{\text{RNN},W^{(k)}}(\mathbf{z}_k)) = \frac{1}{T-d} \left( \sum_{t=d}^{T} \left( \frac{-y_t^{(k)} \cdot e^{-y_t^{(k)} f_t^{\text{RNN},W}(\mathbf{z}_k)}}{1 + e^{-y_t^{(k)} f_t^{\text{RNN},W}(\mathbf{z}_k)}} \right) \cdot \left( \nabla_{W^{(k)}} f_t^{\text{RNN},W^{(k)}}(\mathbf{z}_k) \right) \right) \tag{27}$$

and since $0 \leq \frac{x}{1+x} \leq 1$ for any $x \geq 0$, we conclude that:

$$\left\| \nabla_{W^{(k)}} \ell(\mathbf{y}_k \, , \, f^{\text{RNN},W^{(k)}}(\mathbf{z}_k)) \right\|_F \leq \max_{d \leq t \leq T} \left\| \nabla_{W^{(k)}} f_t^{\text{RNN},W^{(k)}}(\mathbf{z}_k) \right\|_F \tag{28}$$

Now we will use an induction over $i$, to show that $\left\| W^{(i+1)} - W^{(0)} \right\|_F = O\left( \frac{(i+1)T^8}{m \cdot \sqrt{n}} \right)$ for any $0 \leq i < n$. By the induction hypothesis, lemma 10 assure us that for wide enough networks $m^\star = \Omega\left( \max\left\{ T^6 \log^3\left( \frac{n \cdot T}{\delta} \right), \sqrt{n}T^8 \right\} \right)$, with probability of at least $1 - \delta$ over the initialization of $f^{\text{RNN}}$, $\max_{d \leq t \leq T} \left\| \nabla_{W^{(k)}} f_t^{\text{RNN},W^{(k)}}(\mathbf{z}_k) \right\|_F = O\left( T^8 \right)$. Therefore, in this case

$$\left\| W^{(k+1)} - W^{(k)} \right\|_F = \eta O\left( T^8 + \frac{1}{m} \right) = O\left( \frac{T^8}{m \cdot \sqrt{n}} + \frac{1}{m^2 \sqrt{n}} \right) \tag{29}$$

and hence $\left\| W^{(i+1)} - W^{(0)} \right\|_F = O\left( \frac{(i+1)T^8}{m \cdot \sqrt{n}} \right)$ as required.

Now after we showed the assumptions of lemma 8 upholds, we can use it to obtain first order Taylor approximation of the training loss:

$$L_i\left( W^{(i)} \right) - L_i\left( W^\star \right) \leq \left\langle \nabla_{W^{(i)}} \ell(\mathbf{y}_i \, , \, f^{\text{RNN},W^{(i)}}(\mathbf{z}_i)), W^{(i)} - W^\star \right\rangle \tag{30}$$

$$+ \max_{d \leq t \leq T} \underbrace{\left| \frac{y_t \cdot e^{-y_t f^{\text{RNN},W}(\mathbf{z}_i)}}{1 + e^{-y_t f^{\text{RNN},W}(\mathbf{z}_i)}} \right|}_{\leq 1} \cdot O\left( \left( \frac{R}{\sqrt{m}} \right)^{\frac{1}{3}} T^{10} \sqrt{m} \left( \log m \right) \right) \left\| W^{(i)} - W^\star \right\|_F$$

$$\tag{31}$$

Where we assumed that $m^\star > n$ and therefore $\left\| W^{(i)} - W^{(0)} \right\|_F < O\left( \frac{R}{\sqrt{m}} \right)$.

Using algorithm 2 update rule for $W^{(i+1)}$ again (see eq 26), we can use an inequality from in Shalev-Shwartz & Ben-David (2014)'s lemma 14.1 to get that:

$$\sum_{i=1}^{n} \left\langle \nabla_{W^{(i)}} \ell(\mathbf{y}_i \, , \, f^{\text{RNN},W^{(i)}}(\mathbf{z}_i)) + \sigma_i, W^{(i)} - W^\star \right\rangle \tag{32}$$

$$\leq \frac{\left\| W^{(1)} - W^\star \right\|_F^2}{2\eta} + \frac{\eta}{2} \sum_{i=1}^{n} \left\| \nabla_{W^{(i)}} \ell(\mathbf{y}_i \, , \, f^{\text{RNN},W^{(i)}}(\mathbf{z}_i)) + \sigma_i \right\|_F^2 \tag{33}$$

Now combing with Cauchy–Schwarz inequality we have that:

$$\sum_{i=1}^{n}\left(L_i\left(W^{(i)}\right)-L_i\left(W^{\star}\right)\right)\leq\frac{\left\|W^{(1)}-W^{\star}\right\|_F^2}{2\eta}+\frac{\eta}{2}\sum_{i=1}^{n}\left\|\nabla_{W^{(k)}}\ell(\mathbf{y}_k\,,\,f^{\text{RNN},W^{(k)}}(\mathbf{z}_k))+\sigma_k\right\|_F^2 \tag{34}$$

$$+O\left(\left(\frac{R}{\sqrt{m}}\right)^{\frac{1}{3}}T^{10}\sqrt{m}\left(\log m\right)\right)\left(\sum_{i=1}^{n}\left\|W^{(i)}-W^{\star}\right\|_F\right) \tag{35}$$

$$-\sum_{i=1}^{n}\left\langle\sigma_i,W^{(i)}-W^{\star}\right\rangle \tag{36}$$

Substituting the upper bounds from eqs 28, 29 and using the assumption that $R=\Omega\left(T^{16}\right)$ we get that:

$$\sum_{i=1}^{n}\left(L_i\left(W^{(i)}\right)-L_i\left(W^{\star}\right)\right)\leq\frac{O\left(R^2+T^{16}\right)}{2\eta m}+\frac{\eta\cdot n}{2}O\left(T^8+1\right)^2 \tag{37}$$

$$+O\left(\left(\frac{R}{\sqrt{m}}\right)^{\frac{1}{3}}T^{10}n\left(\log m\right)\right)\left(R+nT^8\right)+O\left(m^{-\frac{3}{2}}\left(n^2T^8+nR\right)\right) \tag{38}$$

$$\leq O\left(R^2\sqrt{n}\right)+O\left(\left(\frac{R}{\sqrt{m}}\right)^{\frac{1}{3}}T^{10}n\left(\log m\right)\right)\left(R+nT^8\right)+O\left(m^{-\frac{3}{2}}n^2R\right) \tag{39}$$

To ensure the left hand side is upper bounded by $O\left(R^2\sqrt{n}\right)$ we will chose $m^{\star}$ such that $\frac{m^{\star}}{(\log^3 m^{\star})}>n^{\frac{9}{2}}$, note that, as required, $m^{\star}$ is polynomial in $n,T$. Then for $m>m^{\star}$ we have that $\sum_{i=1}^{n}L_i\left(W^{(i)}\right)\leq O\left(R^2\sqrt{n}\right)$. Therefore,

$$\frac{1}{n}\sum_{i=1}^{n}L_i\left(W^{(i)}\right)\leq O\left(\frac{R^2}{\sqrt{n}}\right) \tag{40}$$

Now, to prove generalization bound we will follow lemma 4.3 in Ji & Telgarsky (2020) and use a martingale Bernstein bound argument. We begin by showing that during the whole training process, our binary cross entropy loss is bounded. Indeed lemma 9 assure us that :

$$\max_{\mathbf{x}\in\{0,1\}^d}\max_{d<t\leq T}\left|f_t^{\text{RNN},W^{(i)}}(\mathbf{z})\right|=O\left(T^{14}\cdot\sqrt{\frac{n}{m}}+T\right)\leq O\left(T^{14}\right) \tag{41}$$

Therefore, their exist a constant $C>0$ such that the binary cross entropy loss is bounded by $\log\left(1+e^{O\left(T^{14}\right)}\right)\leq C\cdot T^{14}$.

Now, we will define a bounded martingle. For any $i\geq 0$, let $s_i$ denote $(\mathbf{x}_i,\mathbf{y}_i)$ and $s_{0,i}$ denote $(s_0,\ldots,s_i)$. Importantly, the quantity

$$\frac{1}{C\cdot T^{14}}\left(\sum_{t<i}\left(\mathbb{E}_{\mathbf{x}}\left[l\left(\mathbf{y},f^{\text{RNN},W^{(t)}}(\mathbf{z})\right)\right]-l\left(\mathbf{y}_t,f^{\text{RNN},W^{(t)}}(\mathbf{z}_t)\right)\right)\right) \tag{42}$$

is a martingal w.r.t the filration $\sigma\left(s_{0,i-1}\right)$. This martingal difference sequence is given by

$$\frac{1}{C\cdot T^{14}}\left(\mathbb{E}_{\mathbf{x}}\left[l\left(\mathbf{y},f^{\text{RNN},W^{(t)}}(\mathbf{z})\right)\right]-l\left(\mathbf{y}_t,f^{\text{RNN},W^{(t)}}(\mathbf{z}_t)\right)\right)\leq 1 \tag{43}$$

Moreover, we have

$$\mathbb{E}_{s_{0,t}}\left[\frac{1}{C^2\cdot T^{28}}\left(\mathbb{E}_{\mathbf{x}}\left[l\left(\mathbf{y},f^{\text{RNN},W^{(t)}}(\mathbf{z})\right)\right]-l\left(\mathbf{y}_t,f^{\text{RNN},W^{(t)}}(\mathbf{z}_t)\right)\right)^2|\sigma\left(s_{0,i-1}\right)\right] \tag{44}$$

$$=\frac{1}{C^2\cdot T^{28}}\left(\mathbb{E}_{s_{0,t}}\left[l\left(\mathbf{y}_t,f^{\text{RNN},W^{(t)}}(\mathbf{z}_t)\right)^2|\sigma\left(s_{0,i-1}\right)\right]-\mathbb{E}_{\mathbf{x}}\left[l\left(\mathbf{y},f^{\text{RNN},W^{(t)}}(\mathbf{z})\right)\right]^2\right) \tag{45}$$

$$\leq\mathbb{E}_{s_{0,t}}\left[\frac{1}{C\cdot T^{14}}l\left(\mathbf{y}_t,f^{\text{RNN},W^{(t)}}(\mathbf{z}_t)\right)|\sigma\left(s_{0,i-1}\right)\right] \tag{46}$$

$$=\frac{1}{C\cdot T^{14}}\cdot\mathbb{E}_{\mathbf{x}}\left[l\left(\mathbf{y},f^{\text{RNN},W^{(t)}}(\mathbf{z})\right)\right] \tag{47}$$

Therefore, by lemma C.2 in Ji & Telgarsky (2020) we have that with probability $1 - \delta$

$$\frac{1}{C \cdot T^{14}} \sum_{i=1}^{n} \left( \mathbb{E}_{\mathbf{x}} \left[ l \left( \mathbf{y}, f^{\text{RNN}, W^{(i)}} (\mathbf{z}) \right) \right] - l \left( \mathbf{y}_i, f^{\text{RNN}, W^{(i)}} (\mathbf{z}_i) \right) \right) \tag{48}$$

$$\leq \frac{1}{C \cdot T^{14}} (e - 2) \cdot \sum_{i=1}^{n} \mathbb{E}_{\mathbf{x}} \left[ l \left( \mathbf{y}, f^{\text{RNN}, W^{(i)}} (\mathbf{z}) \right) \right] + \ln \left( \frac{1}{\delta} \right) \tag{49}$$

And hence

$$\sum_{i=1}^{n} \mathbb{E}_{\mathbf{x}} \left[ l \left( \mathbf{y}, f^{\text{RNN}, W^{(t)}} (\mathbf{z}) \right) \right] \leq \frac{1}{(3 - e)} \sum_{i=1}^{n} l \left( \mathbf{y}_i, f^{\text{RNN}, W^{(i)}} (\mathbf{z}_i) \right) + O \left( T^{14} \right) \ln \left( \frac{1}{\delta} \right) \tag{50}$$

$$= O \left( R^2 \sqrt{n} \right) + O \left( R \ln \left( \frac{1}{\delta} \right) \right) \tag{51}$$

Finally, for $y_t \cdot f_t^{\text{RNN}, W} (\mathbf{z}) < 0$ we have that $\log \left( 1 + e^{-y_t \cdot f_t^{\text{RNN}, W} (\mathbf{z})} \right) > \log 2$. In addition, clearly $\log \left( 1 + e^{-y_t \cdot f_t^{\text{RNN}, W} (\mathbf{z})} \right) > 0$. Therefore, we conclude that $\frac{1}{T - d} \sum_{t=d}^{T} l_{o-1} \left( y_t, f_t^{\text{RNN}, W^{(i)}} (\mathbf{z}) \right) < \frac{1}{\log 2} l \left( \mathbf{y}, f^{\text{RNN}, W^{(i)}} (\mathbf{z}) \right)$ and thus eq 24 uphold. $\qquad \square$

## D  EXTENSION FOR GD WITH FINITE PRECISION

In this section, we prove theorem 2 from the main text still holds when using the gradient descent based algorithm 3, instead of the **stochastic** gradient descent based algorithm 2. . Establishing our positive results that the parities task is efficiently learnable with sub-task decomposition supervision in the exact same setting of the negative results that show that learning is impossible without intermediate supervision, presented in section F. We will follow the proof in section C while taking into account the full non-stochastic gradients.

---

**Algorithm 3:** Training $f^{\text{RNN}}$ with finite precision GD

**Data:** Data set $\mathcal{D}$, learning rate $\eta$, finite precision $\sigma$.
**Initialization:** The entries of $W^{(0)}, A, M_0$ are generated i.i.d. from $N(0, \frac{2}{m})$. The entries of $B$ are generated i.i.d. from $N(0, \frac{1}{m})$.
**for** $i = 1, 2, 3 ... n$ **do**
    Get arbitrary $\sigma$-approximation of the gradient:
    $G^{(i)} \in \mathcal{B}_{\infty}{}^{11} \left( \mathbb{E}_{\mathbf{x}} \left[ \nabla_{W^{(i-1)}} \ell(\mathbf{y}, f^{\text{RNN}, W^{(i-1)}} (\mathbf{z})) \right], \sigma \right)$.
    Update weights:
    $W^{(i)} = W^{(i-1)} - \eta G^{(i)}$.
**end**

---

We begin by sampling a *fake training set* denoted by $\mathbf{x}^{(1)}, \ldots, \mathbf{x}^{(n)}, \mathbf{y}^{(1)}, \ldots, \mathbf{y}^{(n)}$. Essentially, we will apply the same reasoning as in section C with this fake training points. As in the finite precision SGD case it is enough to prove a lemma that is equivalent to lemma 4 in section C.

**Lemma 5.** *Let $n \in \mathbb{N}$, and denote by $L_i (W) := l \left( \mathbf{y}^{(i)}, f^{RNN, W} \left( \mathbf{z}^{(i)} \right) \right)$ the training loss. Suppose there exists $W^{\star} \in \mathcal{B}^{12} \left( W^{(0)}, \frac{R}{\sqrt{m}} \right)$ such that $R = O \left( poly \left( T \right) \right) = \Omega \left( T^{16} \right)$ and $L_i \left( W^{\star} \right) \leq \frac{1 + R^2}{n}$. Then for any $\delta > 0$ there exists $m^{\star} = poly \left( n, R, T, \delta^{-1} \right)$ such that if $m > m^{\star}$ then with probability at least $1 - \delta$ algorithm 3 with $\eta = \frac{1}{m \sqrt{n}}$ and finite precision $\sigma = O \left( \frac{1}{m} \right)$ will output:*

$$\mathbb{E}_{\mathbf{x}} \left[ \left( \frac{1}{n (T - d)} \right) \sum_{t=d}^{T} \sum_{i=1}^{n} l_{0-1} \left( y_t, f_t^{RNN, W^{(i)}} (\mathbf{z}) \right) \right] < O \left( \frac{R^2 + \log \frac{1}{\delta}}{\sqrt{n}} \right) \tag{52}$$

---

[11] The ball is with respect to the distance that defined by the max matrix norm, *i.e.* the elementwise distances are at most $\sigma$.

[12] The ball is with respect to the distance that defined by the Frobenius matrix norm.

*Proof.* We begin by showing that there exists $\tilde{W} \in \mathcal{B}\left(W^{(0)}, \frac{T^8}{\sqrt{m}}\right)$ such that:

$$\mathbb{E}_{\mathbf{x}}\left[l\left(\mathbf{y}, f^{\mathrm{RNN},\tilde{W}}(\mathbf{z})\right)\right] \leq O\left(\frac{R^2}{\sqrt{n}}\right) \tag{53}$$

Indeed, since the minimum can not be larger than the mean, eq'50 in the proof of lemma 4 assure us that under the assumptions of lemma 5, for $m > \max\left\{n^2, \ln^4\left(\frac{1}{\delta}\right)\right\}$, with probability of at least $1 - \delta$ algorithm 2 will reach such $\tilde{W}$ during the first $\max\left\{n, \ln^2\left(\frac{1}{\delta}\right)\right\}$ SGD iteration.

Now we shows that that with high probability over the initialization of $f^{\mathrm{RNN}}$, at any iteration $i \leq n$ of algorithm 3, the distance of the learned hidden weights matrix $W^{(i)}$ from its initialization point $W^{(0)}$ is not too large. As a results we will get that the assumption of lemma 8 uphold, and therefore its upper bound of the deviation from linearization is valid.

By the triangle inequality for any $0 \leq i < n$ we have that:

$$\left\|W^{(i+1)} - W^{(0)}\right\|_F \leq \sum_{k=0}^{i}\left\|W^{(k+1)} - W^{(k)}\right\|_F \tag{54}$$

Substituting algorithm 3 update rule for $W^{(k+1)}$, we get that there exist $\|\sigma_i\|_\infty < \sigma$ such that:

$$W^{(k+1)} - W^{(k)} = -\eta\left(\mathbb{E}_{\mathbf{x}}\left[\nabla_{W^{(k)}}\ell(\mathbf{y}, f^{\mathrm{RNN},W^{(k)}}(\mathbf{z}))\right] + \sigma_k\right) \tag{55}$$

Now, explicitly writing $\nabla_{W^{(k)}}\ell(\mathbf{y}, f^{\mathrm{RNN},W^{(k)}}(\mathbf{z}))$ with the chain rule we have that:

$$\nabla_{W^{(k)}}\ell(\mathbf{y}, f^{\mathrm{RNN},W^{(k)}}(\mathbf{z})) = \frac{1}{T-d}\left(\sum_{t=d}^{T}\left(\frac{-y_t \cdot e^{-y_t f_t^{\mathrm{RNN},W}(\mathbf{z})}}{1 + e^{-y_t f_t^{\mathrm{RNN},W}(\mathbf{z})}}\right) \cdot \left(\nabla_{W^{(k)}}f_t^{\mathrm{RNN},W^{(k)}}(\mathbf{z})\right)\right) \tag{56}$$

and since $0 \leq \frac{x}{1+x} \leq 1$ for any $x \geq 0$, we conclude by Jensen's inequality that:

$$\left\|\mathbb{E}_{\mathbf{x}}\left[\nabla_{W^{(k)}}\ell(\mathbf{y}, f^{\mathrm{RNN},W^{(k)}}(\mathbf{z}))\right]\right\|_F \leq \frac{1}{T-d}\sum_{t=d}^{T}\mathbb{E}_{\mathbf{x}}\left[\left\|\nabla_{W^{(k)}}f_t^{\mathrm{RNN},W^{(k)}}(\mathbf{z})\right\|_F\right] \tag{57}$$

Now we will use an induction over $i$, to show that $\left\|W^{(i+1)} - W^{(0)}\right\|_F = O\left(\frac{(i+1)T^8}{m\cdot\sqrt{n}}\right)$ for any $0 \leq i < n$. By the induction hypothesis, lemma 10 assure us that for wide enough networks $m^\star = \Omega\left(\max\left\{T^7\log^3\left(\frac{n\cdot T}{\delta}\right), \sqrt{n}T^8\right\}\right)$, with probability of at least $1 - \delta$ over the initialization of $f^{\mathrm{RNN}}$, $\mathbb{E}_{\mathbf{x}}\left[\left\|\nabla_{W^{(k)}}f_t^{\mathrm{RNN},W^{(k)}}(\mathbf{z})\right\|_F\right] = O\left(T^8\right)$ for any $d \leq t \leq T$. Therefore, in this case

$$\left\|W^{(k+1)} - W^{(k)}\right\|_F = \eta O\left(T^8 + 1\right) = O\left(\frac{T^8}{m\cdot\sqrt{n}} + \frac{1}{m^2\sqrt{n}}\right) \tag{58}$$

and hence $\left\|W^{(i+1)} - W^{(0)}\right\|_F = O\left(\frac{(i+1)T^8}{m\cdot\sqrt{n}}\right)$ as required.

Now after we showed the assumptions of lemma 11 upholds, we can use it to obtain first order Taylor approximation of the training loss for any $\mathbf{x}$:

$$l\left(\mathbf{y}, f^{\mathrm{RNN},W^{(i)}}(\mathbf{z})\right) - l\left(\mathbf{y}, f^{\mathrm{RNN},\tilde{W}}(\mathbf{z})\right) \leq \left\langle\nabla_{W^{(i)}}\ell(\mathbf{y}, f^{\mathrm{RNN},W^{(i)}}(\mathbf{z})), W^{(i)} - \tilde{W}\right\rangle \tag{59}$$

$$+ \max_{d \leq t \leq T}\underbrace{\left|\frac{y_t \cdot e^{-y_t f^{\mathrm{RNN},W}(\mathbf{z})}}{1 + e^{-y_t f^{\mathrm{RNN},W}(\mathbf{z})}}\right|}_{\leq 1} \cdot O\left(\left(\frac{T^8}{\sqrt{m}}\right)^{\frac{1}{3}}T^{10}\sqrt{m}\left(\log m\right)\right)\left\|W^{(i)} - \tilde{W}\right\|_F \tag{60}$$

Where we assumed that $m^\star > n$ and therefore $\left\|W^{(i)} - W^{(0)}\right\|_F < O\left(\frac{T^8}{\sqrt{m}}\right)$.

Using algorithm 3 update rule for $W^{(k+1)}$ again (see eq 55), we can use an inequality from Shalev-Shwartz & Ben-David (2014)'s section 14.1.1 to get that

$$\sum_{i=1}^{n} \left\langle \mathbb{E}_{\mathbf{x}} \left[ \nabla_{W^{(i)}} \ell(\mathbf{y}, f^{\mathrm{RNN},W^{(i)}}(\mathbf{z})) \right] + \sigma_i, W^{(i)} - \tilde{W} \right\rangle \tag{61}$$

$$\frac{\left\| W^{(1)} - \tilde{W} \right\|_F^2}{2\eta} + \frac{\eta}{2} \sum_{i=1}^{n} \left\| \mathbb{E}_{\mathbf{x}} \left[ \nabla_{W^{(i)}} \ell(\mathbf{y}, f^{\mathrm{RNN},W^{(i)}}(\mathbf{z})) \right] + \sigma_i \right\|_F^2 \tag{62}$$

Now, we can take expectation over eq 59, and combine the Cauchy–Schwarz inequality with the above bound and get that:

$$\sum_{i=1}^{n} \mathbb{E}_{\mathbf{x}} \left[ l \left( \mathbf{y}, f^{\mathrm{RNN},W^{(i)}}(\mathbf{z}) \right) - l \left( \mathbf{y}, f^{\mathrm{RNN},\tilde{W}}(\mathbf{z}) \right) \right] \leq \frac{\left\| W^{(1)} - \tilde{W} \right\|_F^2}{2\eta} \tag{63}$$

$$+ \frac{\eta}{2} \sum_{i=1}^{n} \left\| \mathbb{E}_{\mathbf{x}} \left[ \nabla_{W^{(i)}} \ell(\mathbf{y}, f^{\mathrm{RNN},W^{(i)}}(\mathbf{z})) \right] + \sigma_i \right\|_F^2 + O \left( \left( \frac{T^8}{\sqrt{m}} \right)^{\frac{1}{3}} T^{10} \sqrt{m} \left( \log m \right) \right) \left( \sum_{i=1}^{n} \left\| W^{(i)} - \tilde{W} \right\|_F \right) \tag{64}$$

Substituting the upper bounds from eqs 57, 58 we get that:

$$\sum_{i=1}^{n} \mathbb{E}_{\mathbf{x}} \left[ l \left( \mathbf{y}, f^{\mathrm{RNN},W^{(i)}}(\mathbf{z}) \right) - l \left( \mathbf{y}, f^{\mathrm{RNN},W^{\star}}(\mathbf{z}) \right) \right] \leq \frac{O \left( T^{16} \right)}{2\eta m} + \frac{\eta \cdot n}{2} O \left( T^8 + 1 \right)^2 \tag{65}$$

$$+ O \left( \left( \frac{T^8}{\sqrt{m}} \right)^{\frac{1}{3}} T^{10} n \left( \log m \right) \right) \left( 2nT^8 \right) + O \left( m^{-\frac{3}{2}} \left( n^2 T^8 \right) \right) \tag{66}$$

$$\leq O \left( T^{16} \sqrt{n} \right) + O \left( m^{-\frac{1}{6}} T^{\frac{62}{3}} n^2 \left( \log m \right) \right) + O \left( m^{-\frac{3}{2}} n^2 T^8 \right) \tag{67}$$

To ensure the left hand side is upper bounded by $O \left( R^2 \sqrt{n} \right)$ we will chose $m^{\star}$ such that $\frac{m^{\star}}{(\log^3 m^{\star})} > n^{\frac{9}{2}}$, note that, as required, $m^{\star}$ is polynomial in $n, T$. Then since eq 53 assure us that $\mathbb{E}_{\mathbf{x}} \left[ l \left( \mathbf{y}, f^{\mathrm{RNN},\tilde{W}}(\mathbf{z}) \right) \right] \leq O \left( \frac{R^2}{\sqrt{n}} \right)$, we have that $\sum_{i=1}^{n} \mathbb{E}_{\mathbf{x}} \left[ l \left( \mathbf{y}, f^{\mathrm{RNN},W^{(i)}}(\mathbf{z}) \right) \right] \leq O \left( R^2 \sqrt{n} \right)$ for $m > m^{\star}$. Therefore,

$$\frac{1}{n} \sum_{i=1}^{n} \mathbb{E}_{\mathbf{x}} \left[ l \left( \mathbf{y}, f^{\mathrm{RNN},W^{(i)}}(\mathbf{z}) \right) \right] \leq O \left( \frac{R^2}{\sqrt{n}} \right) \tag{68}$$

Finally, for $y_t \cdot f_t^{\mathrm{RNN},W}(\mathbf{z}) < 0$ we have that $\log \left( 1 + e^{-y_t \cdot f_t^{\mathrm{RNN},W}(\mathbf{z})} \right) > \log 2$. In addition, clearly $\log \left( 1 + e^{-y_t \cdot f_t^{\mathrm{RNN},W}(\mathbf{z})} \right) > 0$. Therefore, we conclude that $\frac{1}{T-d} \sum_{t=d}^{T} l_{o-1} \left( y_t, f_t^{\mathrm{RNN},W^{(i)}}(\mathbf{z}) \right) < \frac{1}{\log 2} l \left( \mathbf{y}, f^{\mathrm{RNN},W^{(i)}}(\mathbf{z}) \right)$ and thus eq 52 uphold.

$\square$

# E LEARNABILITY OF RNNS

In this section we state and extend several lemmas from Wang et al. (2021) and Allen-Zhu et al. (2019). We will use this lemmas in sections C,D for extending theorem 2 in the main text for finite precision SGD and GD.

Following Wang et al. (2021)'s notations, for any target function $h \in \mathcal{H}_{\phi(T, \psi, N)}$ and $n$ samples $\left( \mathbf{z}^{(i)} \right)_{i=1}^{n}$ we will denote:

$$\mathbf{H}_{i,j}^{t} := \frac{1}{m} \left\langle \nabla_{W^{(0)}} f_t^{\mathrm{RNN},W^{(0)}}(\mathbf{z}^{(i)}), \nabla_{W^{(0)}} f_t^{\mathrm{RNN},W^{(0)}}(\mathbf{z}^{(j)}) \right\rangle \tag{69}$$

and

$$\tilde{y}^{(t)} = \begin{pmatrix} h_t\left(\mathbf{z}^{(1)}\right) \\ \vdots \\ h_t\left(\mathbf{z}^{(n)}\right) \end{pmatrix} \tag{70}$$

We start with theorem $4^{13}$ from Wang et al. (2021) measuring the *complexity* of a learned hypothesis class with respect to random initialization of $f^{\text{RNN}}$.

**Lemma 6.** *Let $\delta > 0$ and $n \in \mathbb{N}$, then there exist $m^\star = poly\left(n, \delta^{-1}, T\right)$ such that if $m > m^\star$ then for any $h \in \mathcal{H}_{\phi(T,\psi,N)}$ and $\left(\mathbf{z}^{(i)}\right)_{i=1}^n$ with probability at least $1 - \delta$ over the random initialization of $f^{\text{RNN}}$ there exists matrices $\mathbf{H}^{d,\infty}, \mathbf{H}^{d+1,\infty}, \ldots, \mathbf{H}^{T,\infty} \in \mathbb{R}^{n\times n}$ such that for any $d \le t \le T$ the following holds:*

1. *There exist $\mathbf{v}^{(t)} \in \mathcal{B}_F\left(0, \frac{1}{100\sqrt{\left(\tilde{y}^{(t)}\right)^T \left(\mathbf{H}^{t,\infty}\right)^{-1}\tilde{y}^{(t)}}}\right)$ such that $\mathbf{H}^t + \left(\mathbf{v}^{(t)}\right)^T\mathbf{v}^{(t)} - \mathbf{H}^{t,\infty}$ is positive semi-definite matrix.*

2. $\sqrt{\left(\tilde{y}^{(t)}\right)^T \left(\mathbf{H}^{t,\infty}\right)^{-1}\tilde{y}^{(t)}} = O\left(\sqrt{\phi\left(T, \psi, N\right)}\right).$

Now we state lemma 15 from Wang et al. (2021). Essentially this lemma state that with high probability over the initialization of $f^{\text{RNN}}$, there exists weights that are not far from the initialization and have low training loss.

**Lemma 7.** *Let $\left(\mathbf{z}^{(i)}, \mathbf{y}^{(i)}\right)_{i=1}^n$ be the training set and denote by $L_i\left(W\right) := l\left(\mathbf{y}^{(i)}, f^{\text{RNN},W}\left(\mathbf{z}^{(i)}\right)\right)$ the training loss. Then for any $\delta > 0$ there exists $m^\star = poly\left(n, R, T, \delta^{-1}\right)$ such that if $m > m^\star$ then with probability at least $1 - \delta$, there exist $W^\star \in \mathcal{B}\left(W^{(0)}, \frac{R}{\sqrt{m}}\right)$ such that $L_i\left(W^\star\right) \le \frac{1+R^2}{n}$ and $R = \tilde{O}\left(T\sum_{t=d}^T \sqrt{\left(\tilde{y}^{(t)}\right)^T \left(\mathbf{H}^{t,\infty}\right)^{-1}\tilde{y}^{(t)}}\right).$*

Now we state theorem 13 from Wang et al. (2021). Essentially this lemma bounds the finite width deviation of $f^{\text{RNN}}$ from its infinite width neural tangent kernel (Jacot et al., 2018).

**Lemma 8.** *Let $m, n \in \mathbb{N}$ and $r = O\left(\frac{poly(n,T)}{\sqrt{m}}\right)$, then with probability of at least $1 - O(n)\exp\left(-\Omega\left(m^{\frac{1}{3}}\right)\right)^{14}$ over the initialization of $f^{\text{RNN}}$, for all $\left(\mathbf{z}^{(i)}\right)_{i=1}^n$ and $W, \tilde{W} \in \mathcal{B}\left(W^{(0)}, r\right)$ for any $d \le t \le T$ and $i \in [n]$ the following hold:*

$$\left| f_t^{\text{RNN},\tilde{W}}\left(\mathbf{z}^{(i)}\right) - f_t^{\text{RNN},W}\left(\mathbf{z}^{(i)}\right) - \left\langle \nabla_W f_t^{\text{RNN},W}(\mathbf{z}^{(i)}), \tilde{W} - W \right\rangle \right| \tag{71}$$

$$= O\left(r^{\frac{1}{3}}T^{10}\sqrt{m}\left(\log m\right)\right)\left\|\tilde{W} - W\right\|_F \tag{72}$$

Now we state a simple corollary of lemmas B.3 and C.2.a in Allen-Zhu et al. (2019). Essentially this corollary bound the output of $f^{\text{RNN}}$ with high probability over its initialization.

**Lemma 9.** *Let $m \in \mathbb{N}$ and $W \in \mathcal{B}^{15}\left(W^{(0)}, \frac{poly(T)}{\sqrt{m}}\right)$, then for a given $\mathbf{x}$ and $d < t \le T$ with probability of at least $1 - e^{-\Omega\left(\frac{m^{\frac{1}{3}}}{T^2}\right)}$ over the random initialization of $f^{\text{RNN}}$ the following hold:*

$$\left| f_t^{\text{RNN},W}(\mathbf{z}) \right| = O\left(T^6\left\|W - W^{(0)}\right\|_F + t\right) \tag{73}$$

*Moreover, with probability of at least $1 - e^{O(T)-\Omega\left(\frac{m^{\frac{1}{3}}}{T^2}\right)}$ also the following holds:*

$$\max_{\mathbf{x}\in\{0,1\}^d}\max_{d<t\le T}\left| f_t^{\text{RNN},W}(\mathbf{z}) \right| = O\left(T^6\left\|W - W^{(0)}\right\|_F + T\right) \tag{74}$$

---

[13]Combined with Remark H.1 in Wang et al. (2021)'s supplementary materials.

[14]Note that lemma 13 from Wang et al. (2021) ensure only weaker guaranties of probability $1 - O(n)\exp\left(-\Omega\left(\log m\right)\right)$, but we confirm with the authors that theirs proof proves also our stronger guaranties.

[15]The ball is with respect to the distance that defined by the Frobenius matrix norm.

*Proof.* We begin by showing that it is enough to bound $h_t(\mathbf{z})$. This is indeed the case since it is well known that $\|B\|_2 = O(1)$ with probability of at least $1 - e^{-\Omega(m)}$ (see for example Bandeira & Van Handel (2016)). Now lemma B.3 from Allen-Zhu et al. (2019) assure us that $\left\|h_t^{W^{(0)}}(\mathbf{z})\right\| = O(t)$, and finally lemma C.2.a from Allen-Zhu et al. (2019) bounds the deviation from the initialization $\left\|h_t^W(\mathbf{z}) - h_t^{W^{(0)}}(\mathbf{z})\right\| = O\left(T^6 \left\|W - W^{(0)}\right\|_F\right)$. Finally a simple union bound on all the possible $d$ bits configurations prove also equation 74. □

Now, we use lemma 9 proof together with lemmas B.11 and C.9 from Allen-Zhu et al. (2019) to prove a lemma that bounds the magnitudes of $f^{\text{RNN}}$'s gradients. We will use this lemma for bounding the finite width deviation of $f^{\text{RNN}}$ from its infinite width neural tangent kernel (Jacot et al., 2018). Beyond its application for proving our positive results, lemma 10 below also implies that $f^{\text{RNN}}$, for which we proved our positive results when intermediate supervision exists, upholds the polynomially bounded gradients requirement in the parities negative results with high probability. Thus proving the actual factor enabling efficient learning is, therefore, intermediate supervision. 4

**Lemma 10.** *Let* $m \in \mathbb{N}$ *and* $W \in \mathcal{B}^{16}\left(W^{(0)}, \frac{poly(T)}{\sqrt{m}}\right)$, *then for a given* $\mathbf{x}$ *and* $d < t \leq T$ *with probability of at least* $1 - e^{-\Omega\left(\frac{m^{\frac{1}{3}}}{T^2}\right)}$ *over the random initialization of* $f^{\text{RNN}}$ *the following hold:*

$$\left\|\nabla_W f_t^{RNN,W}(\mathbf{z})\right\|_F = O\left(T^8 \left\|W - W^{(0)}\right\|_F + T^4\right) \tag{75}$$

*Moreover, with probability of at least* $1 - e^{O(T) - \Omega\left(\frac{m^{\frac{1}{3}}}{T^2}\right)}$ *it holds simultaneously for all* $\mathbf{x}$*'s and* $t$*'s.*

*Proof.* Let $D_t \in \mathbb{R}^{m \times m}$ be a diagonal matrix that its diagonal equals to 1 when $h_t^W(\mathbf{z}) > 0$ and otherwise 0. Then we can write the gradients of $f^{\text{RNN}}$ as:

$$\nabla_W f_t^{RNN,W}(\mathbf{z}) = \sum_{i=1}^{t} \prod_{j=t}^{i+1} \left(W^T D_t\right) B D_i h_i^W(\mathbf{z}) \tag{76}$$

The proof of lemma 9 assure us that $\left\|h_t^W(\mathbf{z})\right\| = O\left(T^6 \left\|W - W^{(0)}\right\|_F + t\right)$ with probability of at least $1 - e^{-\Omega\left(\frac{m^{\frac{1}{3}}}{T^2}\right)}$ over the random initialization of $f^{\text{RNN}}$. In addition, lemma B.11 together with lemma C.9.d from Allen-Zhu et al. (2019) assure us that $\left\|\prod_{j=t}^{i+1}\left(W^T D_t\right)\right\|_2 = O\left(T^7 \left\|W - W^{(0)}\right\|_F + T^3\right)$ for all $i, j$ with probability of at least $1 - e^{-\Omega\left(\frac{m^{\frac{1}{3}}}{T^2}\right)}$. Now, it is well known that $\|B\|_2 = O(1)$ with probability of at least $1 - e^{-\Omega(m)}$ (see for example Bandeira & Van Handel (2016)), and clearly $\|D_t\|_2 \leq 1$ for any $t$. Therefore, overall we got that equation 75 holds with probability of at least $1 - e^{-\Omega\left(\frac{m^{\frac{1}{3}}}{T^2}\right)}$. Finally, a simple union bound on all the possible $d$ bits configurations and $T - d$ $t$'s proves the bound holds simultaneously for all $\mathbf{x}$'s and $t$'s. □

Finally, we rely on lemma 8 from Wang et al. (2021) to bounds the finite width deviation of $f^{\text{RNN}}$ from its infinite width neural tangent kernel (Jacot et al., 2018).

**Lemma 11.** *Let* $m \in \mathbb{N}$ *and* $r = O\left(\frac{poly(T)}{\sqrt{m}}\right)$, *then with probability of at least* $1 - \exp\left(O(T \log T) - \Omega\left(m^{\frac{1}{3}}\right)\right)$ *over the initialization of* $f^{\text{RNN}}$, *for all* $W, \tilde{W} \in \mathcal{B}\left(W^{(0)}, r\right)$ *the following hold:*

$$\max_{d \leq t \leq T} \max_{\mathbf{z}} \left(\left|f_t^{RNN,\tilde{W}}(\mathbf{z}) - f_t^{RNN,W}(\mathbf{z}) - \left\langle \nabla_W f_t^{RNN,W}(\mathbf{z}), \tilde{W} - W \right\rangle\right|\right) \tag{77}$$

$$= O\left(r^{\frac{1}{3}} T^{10} \sqrt{m} \left(\log m\right)\right) \left\|\tilde{W} - W\right\|_F \tag{78}$$

*Proof.* Simple union bound on all the possible $d$ bits configurations and $T - d$ $t$'s proves the bound in lemma 8 holds simultaneously for all $\mathbf{x}$'s and $t$'s. □

---

[16]The ball is with respect to the distance that defined by the Frobenius matrix norm.

# F   BIT-SUBSET PARITY END-TO-END NEGATIVE RESULT PROOFS

In this section, we present a theorem stating that without intermediate supervision, for any neural network with gradients that are polynomially bounded, one must use an exponential number of GD steps to achieve non negligible accuracy in the above-presented task of bit subset parity.

**Theorem 4.** *Let $f_\theta$ be any neural-network with $\mathbb{E}_{\mathbf{x}}\left[\left\|\frac{\partial}{\partial\theta}f_\theta\left(\mathbf{x}\right)\right\|^2\right] = O\left(poly\left(d\right)\right)$[17] (polynomially bounded gradients), and let $\mathcal{A}$ be any iterative gradient-based[18] optimization algorithm that runs for $n = O\left(poly\left(d\right)\right)$ iterations, and at each iteration receives $\Omega\left(e^{-d/3}\right)$-approximation of $\mathbb{E}_{\mathbf{x}}\left[\nabla l\left(y, f_\theta\left(\mathbf{x}\right)\right)\right]$. Then, with probability of at least $1 - O\left(e^{-d/3}\right)$ over the target parity function, the loss of $\mathcal{A}$ will be* **higher** *than $\frac{1}{2} - O(e^{-d})$.*

This theorem follows directly from Shalev-Shwartz et al. (2017) and Shamir (2018) and from the fact that random guessing has high zero-one loss, see full proof below. Importantly, lemma 10 shows that $f^{\text{RNN}}$, for which we proved our positive results when intermediate supervision exists, upholds the polynomially bounded gradients requirement in theorem 4 above with high probability. Moreover, section D shows that our positive results holds also for an finite-precision gradient descent optimization algorithm, for with theorem 4 proves the negative results. So overall both the positive and negative proof are on the exact same setup and hence corollary 1follows.

We will use abuse of notations and sometimes identify a predictor $h \in \mathcal{H}$ with the corresponding vector in $\{0,1\}^d$ that his $i$'th coordinate equals to 1 if $i$ is one of the indices in the subset and 0 otherwise. We start by describing a measure from Shalev-Shwartz et al. (2017) that quantifies the amount of "signal" on the underlying target function contained in the gradient. Consider the stochastic optimization problem associated with learning a target function $h$.

$$\min_{\mathbf{w}} F_h\left(\mathbf{w}\right) \tag{79}$$
$$\text{where: } F_h\left(\mathbf{w}\right) := \mathbb{E}_{\mathbf{x}}\left[l\left(p_{\mathbf{w}}\left(x\right), h\left(\mathbf{x}\right)\right)\right]$$

where $l$ is a loss function, $\mathbf{x}$ are the stochastic inputs, and $p_{\mathbf{w}}$ is some predictor parametrized by a parameter vector $\mathbf{w}$ (e.g. a neural network of a certain architecture). We assume that $F$ is differentiable. We measure the *variance* of the gradient of $F$ with respect to $h$, when $h$ is drawn uniformly at random from a collection of candidate target functions $\mathcal{H}$:

$$\text{Var}\left(\mathcal{H}, F, \mathbf{w}\right) := \mathbb{E}_h\left[\left\|\nabla F_h\left(\mathbf{w}\right) - \mathbb{E}_{\tilde{h}}\left[\nabla F_{\tilde{h}}\left(\mathbf{w}\right)\right]\right\|^2\right] \tag{80}$$

To measure meaningful learnability, we will define the expected loss of *random guessing* from $\alpha \in (0,1)$ fraction of the hypothesis class $\mathcal{H}$ as:

$$E^{\mathcal{H},\alpha,l} := \min_{\substack{h \in \mathcal{H} \\ \tilde{\mathcal{H}} \subseteq H, |\tilde{\mathcal{H}}| \geq \alpha \cdot |\mathcal{H}|}} \mathbb{E}_{\tilde{h} \sim \mathcal{U}(\tilde{\mathcal{H}})}\left[\mathbb{E}_{\mathbf{x}}\left[l\left(h\left(\mathbf{x}\right), \tilde{h}\left(\mathbf{x}\right)\right)\right]\right] \tag{81}$$

As $\alpha$ approaches 1, this measure reflects the expected loss to be attained by randomly assigning a hypothesis from $\mathcal{H}$. We will use this measure in the following lemma, which is a direct corollary of theorem 4 in Shamir (2018). Essentially, this addresses any iterative algorithm (possibly randomized), which relies on an $\epsilon$-approximate gradient oracle to optimize $F_h$ in eq 79. The lemma states that if the number of iterations is not larger than $\text{Var}\left(\mathcal{H}, F, \mathbf{w}\right)^{-1/3}$ then with high probability, the algorithm will return the same predictor independent of $h$.

**Lemma 12.** *Define $\epsilon = \sqrt[3]{\sup_{\mathbf{w}} Var\left(\mathcal{H}, F, \mathbf{w}\right)}$ and let $\mathcal{A}$ be any iterative gradient-based[19] optimization algorithm that runs for $n$ iterations, and at each iteration receives $\Omega\left(\epsilon\right)$-approximation of $\nabla F_h\left(w\right)$. Then the following holds:*

$$P_{h \sim \mathcal{U}(\mathcal{H})}\left[\mathbb{E}_{\mathbf{x}}\left[l\left(h\left(\mathbf{x}\right), p_{\mathcal{A}}\left(\mathbf{x}\right)\right)\right] \geq E^{\mathcal{H}, 1-n\epsilon, l}\right] \geq 1 - n\epsilon \tag{82}$$

---

[17] For any $\theta$ that is reachable by $\mathcal{A}$ with $n$ iterations.

[18] See footnote 3.

[19] See footnote 3.

Lemma 12 above assures us that in order to prove negative learning results regarding some task, it is enough to show it has both exponentially low variance of gradient and high random guessing error. Now we will show both of these for the task of learning bit subset parity presented in section 5. Following Shalev-Shwartz et al. (2017), we start by showing that different parities functions are uncorrelated, and hence this task has exponentially low gradient variance:

**Lemma 13.** *For any* $(i_j)_{j=1}^{\frac{d}{2}} \neq (\tilde{i}_j)_{j=1}^{\frac{d}{2}}$ *we have that:*

$$\mathbb{E}_{\mathbf{x}}\left[(-1)^{\sum_{j=1}^{\frac{d}{2}} x_{i_j}}(-1)^{\sum_{j=1}^{\frac{d}{2}} x_{\tilde{i}_j}}\right] = 0 \tag{83}$$

*Proof.* Denote by $h \neq \tilde{h} \in \{0,1\}^d$ the vectors that corresponding to the $(i_j)_{j=1}^{\frac{d}{2}}, (\tilde{i}_j)_{j=1}^{\frac{d}{2}}$. Then

$$\mathbb{E}_{\mathbf{x}}\left[(-1)^{\sum_{j=1}^{\frac{d}{2}} x_{i_j}}(-1)^{\sum_{j=1}^{\frac{d}{2}} x_{\tilde{i}_j}}\right] = \mathbb{E}_{\mathbf{x}}\left[(-1)^{\langle \mathbf{x},h \rangle}(-1)^{\langle \mathbf{x},\tilde{h} \rangle}\right] \tag{84}$$

$$= \mathbb{E}_{\mathbf{x}}\left[(-1)^{\langle \mathbf{x},h+\tilde{h} \rangle}\right] \tag{85}$$

$$= \mathbb{E}_{\mathbf{x}}\left[\prod_{j=1}^{d}(-1)^{x_j(h_j+\tilde{h}_j)}\right] \tag{86}$$

Since the coordinates are independent we can swap the order of the expectation and the product:

$$= \prod_{j=1}^{d} \mathbb{E}_{\mathbf{x}}\left[(-1)^{x_j(h_j+\tilde{h}_j)}\right] = \prod_{j=1}^{d} \frac{(-1)^{h_j+\tilde{h}_j} + (-1)^0}{2} \tag{87}$$

$$= \prod_{j=1}^{d} \frac{(-1)^{h_j+\tilde{h}_j} + 1}{2} \tag{88}$$

Finally, since $h \neq \tilde{h}$ there exists some $j \in [d]$ such that $h_j \neq \tilde{h}_j$ and therefore $h_j + \tilde{h}_j = 1$ and $\prod_{j=1}^{d} \frac{(-1)^{h_j+\tilde{h}_j}+1}{2} = 0.$ $\square$

Now following Shalev-Shwartz et al. (2017), we show that the zero correlation between different parities implies exponentially low gradient variance.

**Lemma 14.** *Assuming that* $\mathbb{E}_{\mathbf{x}}\left[\left\|\frac{\partial}{\partial \mathbf{w}} p_{\mathbf{w}}(\mathbf{x})\right\|^2\right] = O\left(poly\left(d\right)\right)$, *then* $\sup_{\mathbf{w}} Var\left(\mathcal{H}, F, \mathbf{w}\right) = O\left(|\mathcal{H}|^{-1}\right)$ *where* $\mathcal{H}$ *denote the hypothesis class of $d$ dimensional parities described in section 5.*

*Proof.* We use theorem 1 from Shalev-Shwartz et al. (2017). Essentially, this theorem shows that if the functions in the hypothesis class are uncorrelated, then the variance is upper bounded by $\mathbb{E}_{\mathbf{x}}\left[\left\|\frac{\partial}{\partial \mathbf{w}} p_{\mathbf{w}}(\mathbf{x})\right\|^2\right]$ times the inverse of the hypothesis class size.

In our case, lemma 13 above shows that:

$$(i_j)_{j=1}^{\frac{d}{2}} \neq (\tilde{i}_j)_{j=1}^{\frac{d}{2}} \implies \mathbb{E}_{\mathbf{x}}\left[(-1)^{\sum_{j=1}^{\frac{d}{2}} x_{i_j}}(-1)^{\sum_{j=1}^{\frac{d}{2}} x_{\tilde{i}_j}}\right] = 0 \tag{89}$$

And therefore we have that

$$Var\left(\mathcal{H}, F, \mathbf{w}\right) \leq |\mathcal{H}|^{-1} \mathbb{E}_{\mathbf{x}}\left[\left\|\frac{\partial}{\partial \mathbf{w}} p_{\mathbf{w}}(\mathbf{x})\right\|^2\right]. \tag{90}$$

. $\square$

Having low gradient variance alone does not imply that learning is impossible, it only implies that all hypotheses perform similarly. To show the negative result for our parity problem, we must also show that the random guessing error is high. Lemma 15 establishes this, showing that the task of learning bit-subset parity has non-vanishing random guessing error of $\Theta(1)$ with respect to the dimension $d$, and linear with respect to the fraction of the hypothesis class from which the guessing occurs.

**Lemma 15.** *Let $\mathcal{H}$ denote the hypothesis class of $d$ dimensional parities described in section 5, and $l_{0-1}$ denote the zero one loss. then the following holds:*

$$\forall \alpha \in (0,1) \quad E^{\mathcal{H},\alpha,l_{0-1}} = \frac{1}{2}\left(1 - \frac{1}{\alpha|\mathcal{H}|}\right) \tag{91}$$

*Proof.* Let $h \in \mathcal{H}$, we begin by writing the zero-one loss with elementary algebraic operation:

$$\mathbb{E}_{\tilde{h}\sim\mathcal{U}(\tilde{\mathcal{H}})}\left[\mathbb{E}_{\mathbf{x}}\left[l_{0-1}\left(h(\mathbf{x}),\tilde{h}(\mathbf{x})\right)\right]\right] = \mathbb{E}_{\tilde{h}\sim\mathcal{U}(\tilde{\mathcal{H}}\subseteq\{0,1\}^d)}\left[\mathbb{E}_{\mathbf{x}}1_{(-1)^{\langle\mathbf{x},h\rangle}\neq(-1)^{\langle\mathbf{x},\tilde{h}\rangle}}\right] \tag{92}$$

$$= \mathbb{E}_{\tilde{h}\sim\mathcal{U}(\tilde{\mathcal{H}})}\left[\mathbb{E}_{\mathbf{x}}\left(\frac{1-(-1)^{\langle\mathbf{x},h\rangle}\cdot(-1)^{\langle\mathbf{x},\tilde{h}\rangle}}{2}\right)\right] \tag{93}$$

$$= \frac{1}{2}\left(1 - \mathbb{E}_{\tilde{h}\sim\mathcal{U}(\tilde{\mathcal{H}})}\mathbb{E}_{\mathbf{x}}\left((-1)^{\langle\mathbf{x},h+\tilde{h}\rangle}\right)\right) \tag{94}$$

Since the coordinates are independent we can swap the order of the expectation and the product:

$$= \frac{1}{2}\left(1 - \mathbb{E}_{\tilde{h}\sim\mathcal{U}(\tilde{\mathcal{H}})}\left(\prod_{i=1}^{d}\mathbb{E}_{x_i}(-1)^{x_i(h_i+\tilde{h}_i)}\right)\right) \tag{95}$$

$$= \frac{1}{2}\left(1 - \frac{1}{2^d}\mathbb{E}_{\tilde{h}\sim\mathcal{U}(\tilde{\mathcal{H}})}\left[\prod_{i=1}^{d}\left[(-1)^{h_i+\tilde{h}_i}+(-1)^0\right]\right]\right) \tag{96}$$

Now, in cases where $h \neq \tilde{h}$ there exists some $j \in [d]$ such that $h_j \neq \tilde{h}_j$ . Therefore, in such cases $h_j + \tilde{h}_j = 1$ which implies that $\prod_{j=1}^{d}\frac{(-1)^{h_j+\tilde{h}_j}+1}{2} = 0$ and we get that:

$$= \frac{1}{2}\left(1 - \frac{1}{2^d}\mathbb{E}_{\tilde{h}\sim\mathcal{U}(\tilde{\mathcal{H}})}\left[2^d\cdot 1_{h=\tilde{h}}\right]\right) \tag{97}$$

$$= \frac{1}{2}\left(1 - \mathbb{E}_{\tilde{h}\sim\mathcal{U}(\tilde{\mathcal{H}})}\left[1_{h=\tilde{h}}\right]\right) \tag{98}$$

Now clearly

$$\mathbb{E}_{\tilde{h}\sim\mathcal{U}(\tilde{\mathcal{H}})}\left[1_{h=\tilde{h}}\right] = \begin{cases} \frac{1}{|\tilde{\mathcal{H}}|} & h \in \tilde{\mathcal{H}} \\ 0 & h \notin \tilde{\mathcal{H}} \end{cases} \tag{99}$$

And therefore

$$\mathbb{E}_{\tilde{h}\sim\mathcal{U}(\tilde{\mathcal{H}})}\left[\mathbb{E}_{\mathbf{x}}\left[l_{0-1}\left(h(\mathbf{x}),\tilde{h}(\mathbf{x})\right)\right]\right] = \frac{1}{2} - \frac{1}{2}\cdot\begin{cases} \frac{1}{|\tilde{\mathcal{H}}|} & h \in \tilde{\mathcal{H}} \\ 0 & h \notin \tilde{\mathcal{H}} \end{cases} \tag{100}$$

$\square$

Finally, by combining lemmas 14,15 with lemma 12 we prove theorem 4.

*Proof.* By lemma 14 we know that $\sup_{\mathbf{w}} \text{Var}(\mathcal{H}, F, \mathbf{w}) = O\left(|\mathcal{H}|^{-1}\right)$. Now since $\binom{n}{k} > \left(\frac{n}{k}\right)^k$ for any $k < n$, we have that $|\mathcal{H}| = \binom{d}{\frac{d}{2}} = \Omega\left(e^d\right)$ *i.e.* we have exponentially low variance. Therefore, lemma 12 assure us that with probability of at least $1 - n \cdot O\left(e^{-d/3}\right)$ the loss of $\mathcal{A}$ will be higher than $E^{\mathcal{H},1-n\cdot O\left(|\mathcal{H}|^{-\frac{1}{3}}\right),l_{0-1}}$. Finally, lemma 15 implies that:

$$E^{\mathcal{H},1-n\cdot O\left(|\mathcal{H}|^{-\frac{1}{3}}\right),l_{0-1}} = \frac{1}{2}\left(1 - \frac{1}{|\mathcal{H}| - \max\left\{O\left(|\mathcal{H}|^{\frac{2}{3}}\right)\cdot n, |\mathcal{H}|-1\right\}}\right) \tag{101}$$

$\square$

Table 1: Learning Bit-Subset Parity with Transformers results. We say that a model is better than random when its validation accuracy are higher than $60\%$. In addition, for each task we also evaluated the test accuracy after $100k$ training iteration. The reported test accuracies are of the models with the best hyper-parameters according to the validation loss. Each value after the $\pm$ sign indicates the two standard deviations over three different random seeds.

| Task Size | With Intermediate Supervision | | Without Intermediate Supervision | |
| --- | --- | --- | --- | --- |
| | Test Accuracy | Iterations until better than random | Test Accuracy | Iterations until better than random |
| 8 Bits | $100\% \pm 0.0$ | $79 \pm 21.16$ | $100\% \pm 0.0$ | $185 \pm 79.89$ |
| 10 Bits | | | $100\% \pm 0.0$ | $843 \pm 424.22$ |
| 12 Bits | | | $100\% \pm 0.0$ | $4.049\text{K} \pm 1.132\text{K}$ |
| 14 Bits | | | $100\% \pm 0.0$ | $31.16\text{K} \pm 7.23\text{K}$ |
| 16 Bits | $100\% \pm 0.0$ | $333 \pm 67.002$ | $51.82\% \pm 1.06\%$ | $211.66\text{K} \pm 57.9\text{K}$ |
| 18 Bits | | | $50.16\% \pm 2.65\%$ | $1.534\text{M} \pm 1.094\text{M}$ |
| 32 Bits | $100\% \pm 0.0$ | $1.48\text{K} \pm 61.101$ | $49.87\% \pm 0.72\%$ | $> 100\text{K}$ |
| 64 Bits | $100\% \pm 0.0$ | $11\text{K} \pm 1.154\text{K}$ | $49.93\% \pm 0.43\%$ | $> 100\text{K}$ |
| 128 Bits | $99.96\% \pm 0.06\%$ | $58.33\text{K} \pm 5.20\text{K}$ | $50.71\% \pm 1.26\%$ | $> 100\text{K}$ |
| 256 Bits | $48.24\% \pm 2.06\%$ | $720.16\text{K} \pm 83.17\text{K}$ | $50.34\% \pm 0.73\%$ | $> 100\text{K}$ |

Table 2: Hyper-Parameters and the random seed that examined in the experiment of learning bit-subset parity with Transformers.

| | |
| --- | --- |
| Learning Rate | $10^{-6}, 10^{-5}, 10^{-4}$ |
| Weight Decay | $0, 10^{-6}, 10^{-4}, 10^{-2}$ |
| Dropout | 0 |
| Batch Size | 32 |
| Warmup Steps | $1k$[20] |
| Total Steps | $100k$[21] |
| Number of Layers | 12 |
| Hidden Size | 768 |
| FFN Inner Hidden Size | 3072 |
| Initializer Range | 0.02 |
| Adam $\beta_1$ | 0.9 |
| Adam $\beta_2$ | 0.999 |
| Adam $\epsilon$ | $10^{-8}$ |
| Validation Size | $\min(1024, 12.5\%$ of the data$)$ |
| Test Size | $\min(1024, 12.5\%$ of the data$)$ |
| Dataset Seed | 27, 67, 93 |

## G  BIT-SUBSET PARITY EXPERIMENTS FULL DETAILS

Section 5.2 in the main text empirically demonstrates that there exists a large gap when using the commonly used Transformer architecture for learning bit subset parity with and without intermediate supervision. In this section, we present the full details of the experiments.

Following the bit-subset parity definition in section 5, we randomly sampled a subset of $d/2$ predefined unique indices and then we randomly sampled non-overlapping training, validation and test datasets. For reproducibility purposes, Table 2 reports the random seeds that used for this sampling. Then, we used the standard implementation of GPT-2 from the transformers framework (Wolf et al., 2020) and trained a BERT-Base size Transformer decoder model with a vocabulary size of 2. Even though the small vocabulary size of our model may indicate that it is too shallow for its size (Wies

---

[21] For models that converged after less than $5K$ steps we disabled the learning rate warmup.

[21] For models that did not converged after $100K$ steps we continue the training until convergence. Note that even for thus models Table 1 reports the test accuracy at step $100k$.

et al., 2021), we still choose to use the standard Transformer-Base configuration. See full architecture and optimization details in Table 2. While the evaluation of a model without intermediate supervision is straightforward, we can simply use $argmax$ over the last token logits. This is no longer the case when intermediate supervision is used since at evaluation time the model must create its own intermediate steps. As a result, in this case, we need a decoding algorithm that iteratively predicates the intermediate steps. Since we found in preliminary experiments that greedy decoding tends to be more efficient for longer sequences than random sampling, we choose to use it.

To complement Figure 3 in the main text, Table 1 shows the first iteration[22] for which the model is better than random as well as the test accuracy after $100k$ training iterations. Though "better than random" could be defined in a variety of ways. Figure 4 illustrates a typical learning trajectory and demonstrate the insignificant differences in definitions due to the observed grokking phenomenon (Power et al., 2021). For each task we repeated the task 3 times with different dataset seeds, and perform grid search over the hyper-parameters in Table 2. Table 1 reports the mean and standard deviation test accuracy of the best hyper-parameters according to the validation binary cross-entropy loss, where we break ties by choosing the model that converged faster. As excepted, without intermediate supervision, the Transformer models can not learn beyond random guessing tasks with more than 18 bits. In contrast, with intermediate supervision, the test accuracy is almost perfect even at 128 bits.

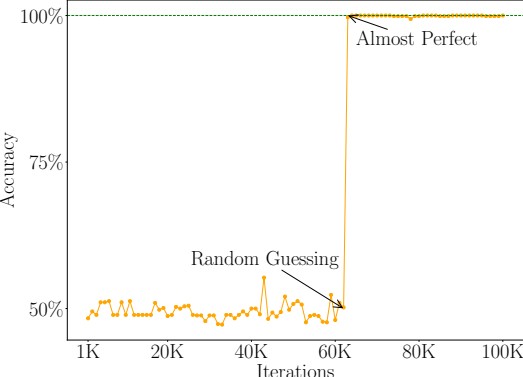

Figure 4: Illustration of a typical bit-subset parity learning trajectory. In these tasks we observed a grokking phenomenon (Power et al., 2021) where very soon after the validation accuracy became higher than random level it also became almost flawless (accuracy > 95%). Therefore, successful learning is not sensitive to the exact cutoff.

---

[22]Validation accuracy was evaluated at 1K-step intervals, except for models that converged in less than 4K steps for which we run the evaluation every 10 steps.

