# OpenReview forum: "Sub-Task Decomposition Enables Learning in Sequence to Sequence Tasks"
_ICLR.cc/2023/Conference — ICLR 2023 poster_

### Official Review · Reviewer_iwZz · 2022-10-22

**Confidence:** 2
**Correctness:** 4
**Technical Novelty And Significance:** 3
**Empirical Novelty And Significance:** 2
**Recommendation:** 5

**Clarity, Quality, Novelty And Reproducibility:**

Clarity: Good
Quality: Ok
Novelty: Ok
Reproducibility: Good

**Strength And Weaknesses:**

Strengthes:
1. Decomposing complexe reasoning tasks into subtasks is currently very popular in NLP, and some theoretically analysis of this paradigm can be helpful.
2. The paper is overall well written and the proofs/theorems looks good to me.

Weaknesses:
1. The authors motivate their research problem by sub-task decomposition of LLMs. However, most current practice of using sub-task decomposition with LLMs are using in-context learning without gradient updates, which is very dissimilar with the settings analyzed in the paper.
2. The model architecture analyzed in the paper is elmon RNN, which is also very different in nature compared with Transformer models (i.e., sequential computation in RNN v.s. parallel encoding with Transformers). The paper could be made more interesting if the proof can be generalized to Transformer architectures.

**Summary Of The Paper:**

This paper theoretically proves that sub-task decomposition can facilitate training of an elmon RNN by reducing the number of gradient updates to polynomial order while undecomposed tasks can't be learned with polynomial updates.

**Summary Of The Review:**

The paper provides some interesting theoretical analysis of the usefulness of sub-task decomposition for seq2seq tasks. However, the settings are somewhat oversimplified and deviate significantly from the main motivation of the paper (sub-task decomposition for NLP/reasoning tasks with LLMs.)

---

> ### Author Response · Authors · 2022-11-18
> **Official Response**
>
> We thank you for your thoughtful feedback.
>
> 1.
>
> The original chain of thought prompting method, which is based on in-context learning, is indeed a prominent example of sub-task decomposition in LLMs. However, training (fine-tuning) language models with the sub-task decomposition signal  is also quite common [1-6], and in fact, it was developed before and used as an inspiration for the chain of thought prompting technique [7].
> That being said, from a theoretical point of view, to prove the exponential gap one needs to prove both negative results on learning without intermediate supervision and positive results for learning with intermediate supervision. While the negative results for our concrete demonstration of bit subset parity (Section 5) rely heavily on “gradient updates” and do not apply to "in-context" learning, our more general negative results (Section 6) can be applied to any polynomial time learning algorithm without intermediate supervision, and hence are valid for "in-context" learning as well. So the only missing part is a positive result for "in-context" learning learnability. Note that in context learning learnability is a more general (and highly interesting) open question, namely, we are not aware of any positive result about why and how LMs learn in context without gradient updates.
> As we said, our analyzed framework of subtask decomposition which includes gradient updates is common in the empirical literature, but we take your point and we will refer to learnability of in context learning (without gradients) as a highly interesting open question in our final version.
>
> 2.
>
> As in the previous point, from a theoretical point of view, to prove the exponential gap one needs to prove both negative results on learning without intermediate supervision and positive results for learning with intermediate supervision.
> Since the negative results generalize for any architecture for which the mild assumption of polynomial bounded gradients uphold, one only needs to prove positive results for the Transformer architecture.
>
> These results are likely valid for many widely-used architectures; indeed, as we show, even the much weaker classical Elman RNN can provably learn this task. Given our result, it is likely that the stronger Transformer can also learn this task, but analyzing and proving positive learnability results for Transformers is challenging. To this end, showing via experiments that Transformer architectures easily learn the bit subset parity task with intermediate supervision, and fail to do so without it, bridges much of the above conceptual gap.
>
> [1] Nazneen Fatema Rajani, Bryan McCann, Caiming Xiong, and Richard Socher. Explain yourself! leveraging language models for commonsense reasoning. In Proceedings of the 57th Annual Meeting of the Association for Computational Linguistics, pp. 4932–4942, Florence, Italy, July 2019. Association for Computational Linguistics. doi: 10.18653/v1/P19-1487. URL https://aclanthology.org/P19-1487.
>
> [2] Karl Cobbe, Vineet Kosaraju, Mohammad Bavarian, Mark Chen, Heewoo Jun, Lukasz Kaiser,Matthias Plappert, Jerry Tworek, Jacob Hilton, Reiichiro Nakano, Christopher Hesse, and John Schulman. Training verifiers to solve math word problems. arXiv preprint arXiv:2110.14168, 2021.
>
> [3] Piotr Pi˛ekos, Mateusz Malinowski, and Henryk Michalewski. Measuring and improving BERT’s mathematical abilities by predicting the order of reasoning. In Proceedings of the 59th Annual Meeting of the Association for Computational Linguistics and the 11th International Joint Conference on Natural Language Processing (Volume 2: Short Papers), pp. 383–394, Online, August 2021. Association for Computational Linguistics. doi: 10.18653/v1/2021.acl-short.49. URL https://aclanthology.org/2021.acl-short.49.
>
> [4] Gabriel Recchia. Teaching autoregressive language models complex tasks by demonstration. arXiv preprint arXiv:2109.02102, 2021.
>
> [5] Maxwell Nye, Anders Johan Andreassen, Guy Gur-Ari, Henryk Michalewski, Jacob Austin, David Bieber, David Dohan, Aitor Lewkowycz, Maarten Bosma, David Luan, Charles Sutton, and Augustus Odena. Show your work: Scratchpads for intermediate computation with language models. In Deep Learning for Code Workshop, 2022. URL https://openreview.net/forum?id=HBlx2idbkbq
>
> [6] Eric Zelikman, Yuhuai Wu, and Noah D. Goodman. Star: Bootstrapping reasoning with reasoning, 2022. URL https://arxiv.org/abs/2203.14465
>
> [7] Jason Wei, Xuezhi Wang, Dale Schuurmans, Maarten Bosma, Ed Chi, Quoc Le, and Denny Zhou.
> Chain of thought prompting elicits reasoning in large language models, 2022.

---

### Official Review · Reviewer_G5w1 · 2022-10-25

**Confidence:** 3
**Clarity, Quality, Novelty And Reproducibility:** The paper is written clearly, is high…
**Correctness:** 4
**Technical Novelty And Significance:** 3
**Empirical Novelty And Significance:** Not applicable
**Recommendation:** 8

**Strength And Weaknesses:**

Strengths
-Well written and interesting.
-Studies a phenomenon that is of great interest to the community right now.

Weaknesses
-The theory is interesting but does not lead to any new methods.

**Summary Of The Paper:**

The paper studies how intermediate supervision (i.e. breaking a task down into steps) can theoretically increase the power of learning complex tasks.

The authors give a real world motivating example of math problems i.e.
"John and I split 12 apples between us, and he gave 1 to his son. How many apples does he have now?" (5)

which can be broken down into:

1. John and I split 12 apples between us. How many apples does he have now (6)
2. John had 6 apples and he gave 1 to his son. How many apples does he have now? (5)

This has been empirically observed in the past and has gained recent prominence due to impressive results with large language models (i.e. chain of thought prompting)

The authors assume a particular model (a simple RNN) and theoretically prove the following:

There exists a binary classification problem of size d such that with intermediate supervision it is possible for seq2seq model to get an arbitrary low error with a polynomial number of gradient updates in d. However, without intermediate supervision, then for any polynomial time learning algorithm the error will be higher than 1/2 - epsilon.

They then explore the specific case of bit subset parity, showing learning curves for BERT that demonstrate that without intermediate supervision the number of training steps required grows very rapidly as the size of the problem increases, whereas with intermediate supervision the number of training steps grows at a much more favorable rate.


**Summary Of The Review:**

I support accepting the paper. It is well motivated, well-written and sheds theoretical insight onto an interesting and relevant problem.

---

> ### Author Response · Authors · 2022-11-18
> **Official Response**
>
> We thank you for your thoughtful and supportive feedback. Regarding inspirations from this paper please see our comment for reviewer T2bv.   We emphasize that while our paper does not propose a novel NLP technique, we believe that a solid theoretical foundation for sub-task decomposition is useful as it principally encourages the curation of datasets with intermediate steps for various NLP tasks. Though chain of thought / multihop works are gaining popularity, there is still a prevalent belief that bigger and stronger networks can crack harder and harder tasks. We think that a theoretical result such as ours can bring some principled arguments into the discussion on what can we hope to see solved by larger networks and what needs to be handled differently (by finding ways to insert intermediate supervision).

---

### Official Review · Reviewer_T2bv · 2022-10-28

**Confidence:** 3
**Correctness:** 4
**Technical Novelty And Significance:** 4
**Empirical Novelty And Significance:** 3
**Recommendation:** 8

**Clarity, Quality, Novelty And Reproducibility:**

In general the paper is quite clear, of high-quality, novel, and reproducible (full proofs provided, source-code for experiments is provided).

**Strength And Weaknesses:**

Strengths
- I was impressed by the generality of their positive results: that any function in P is learnable when suitable sub-task supervision is provided. Their use of boolean circuits to prove this result was very clever.
- Their use of the bit-subset parity problem throughout the paper was effective at making their theoretical result more concrete and easier to digest.
- Although the primary results are theoretical, I do feel that these results have important empirical implications regarding the limits of sequence-to-sequence models in solving complicated tasks that require multiple intermediate reasoning steps, when intermediate sub-task supervision is not provided. It also serves as encouragement for empirical work that provides this intermediate supervision to large language models.

Weaknesses
- It is not immediately clear how the theoretical constructions presented in this paper can be applied to solve real problems in NLP/ML. Perhaps the authors could comment further on how they believe intermediate supervision can be provided in these practical settings.
- NIT: I felt that the clarity could be improved in a few places. For example, I felt that the hypothesis class defined by equation (6) could have been more carefully explained (for example, what is e_z, and what is w?).



**Summary Of The Paper:**

This paper analyzes the learnability of sequence-to-sequence tasks, based on whether or not the answers to sub-tasks are provided as part of the input sequence. It proves a rather general result: That *any* function in the P time complexity class is efficiently learnable by a neural network when sub-task supervision is provided, but there exists functions in P which are not learnable (in polynomial time) without sub-task supervision.

The paper gives significant attention to the “bit-subset parity” problem: this is the problem of determining whether a bit-string of length d has an even or odd number of ones in an unknown subset of d/2 indices. It shows that this problem is efficiently learnable by an RNN when sub-task supervision is provided, but is not learnable (under standard cryptographic or computational hardness assumptions) in polynomial time without this supervision. In this case, the sub-task supervision is the parity of pairs of the unknown indices, recursively followed by the parity of these sub-task parity answers in a binary-tree like structure: $par(x_i1, x_i2), par(x_i3, x_i4), …, par(par(x_i1, x_i2), par(x_i3, x_i4)), …$ (see Figure 2, Section 5). The paper then uses the hardness results regarding this bit-subset parity problem from Shalev-Shwartz et al. (2017) to prove it is not efficiently learnable.

Although the primarily contributions of this paper are theoretical, this paper also provides experiments showing that a powerful transformer network (BERT-base architecture) is unable to learn the bit-subset parity problem for $d \geq 32$ (within ~2M gradient steps), but can easily learn it when the sub-task supervision is provided (e.g., can learn $d=128$ in  $< 100k$ steps).


**Summary Of The Review:**

This paper provides important theoretical results regarding the learnability (or unlearnability) of sequence-to-sequence tasks by neural networks, depending on whether or not sub-task supervision is provided. These results are quite general, and provide important insights for future empirical research in sequence to sequence learning.  As a result, I definitely recommend acceptance for this paper (although my review is only medium confidence, given that I am not super familiar with the related work).

---

> ### Author Response · Authors · 2022-11-18
> **Official Response**
>
> We thank you for your thoughtful and supportive feedback.
>
> 1.
> Regarding the application of the theoretical constructions in the paper for solving real problems in NLP/ML, we believe that our theoretical foundation for sub-task decomposition should encourage the curation of datasets with intermediate steps for various NLP tasks. A clear theoretical result such as ours can make NLP practitioners more mindful of the price paid for trying to teach a network multihop functions without explicit intermediate steps. In addition, the negative results in our concrete demonstration highlight the lack of "divide and conquer" inductive bias in standard neural networks training. In order to circumvent this limitation, one interesting idea is to inject such an inductive bias into the neural network architecture. This is done for example in [1] in a reinforcement learning setting.
>
> 2.
> We thank you for pointing out unclear notations in our presentation, and we have added explanations for symbols that previously appeared in equations without proper explanations. Essentially, the hypothesis class that is defined by equation (6) is a sequence-to-sequence function whose outputs are the signs of multivariable polynomials. Specifically, we assume that each output is a function of at most N  input locations. So we denote the indices of these locations as $j_1, \dots, j_N$., and $z_{j_1}, \dots, z_{j_N}$ denote the sequence tokens at these locations. Now these tokens are either 0 or 1, so $\mathbf e_z$ stands for either $\mathbf e_0$ or $\mathbf e_1$ which are the standard basis of $\mathbb{R}^2$. Finally, for any t $\mathbf{w}^{\left(t\right)}$ denotes a weight vector in $\mathbb{R}^{2N}$. Hence equation (6) requires that the output of each function is the sign of a polynomial activation of the inner product between $\mathbf{w}^{\left(t\right)}$ and the concatenation of N different one-hot input vectors $ \mathbf e_{z_{j_1}}, \dots, \mathbf e_{z_{j_N}}$ .
>
> [1] Yuchen Lu et al. “Learning Task Decomposition with Ordered Memory Policy Network”. In: International Conference on Learning Representations. 2021. url: https://openreview.net/forum?id=vcopnwZ7bC.

---

### Official Review · Reviewer_f3DQ · 2022-10-29

**Confidence:** 2
**Correctness:** 3
**Technical Novelty And Significance:** 3
**Empirical Novelty And Significance:** 3
**Recommendation:** 6

**Clarity, Quality, Novelty And Reproducibility:**

* Clarity: somewhat clear. The paper is easy to follow, but some important details are missing or unclear.
* Quality: somewhat technically sound.
* Novelty: somewhat good. The inspriation is limited.
* Reproducibility: key resources (e.g., proofs) are available and it would be better if sufficient details (e.g., experimental setup) are described such that an expert should be able to reproduce the main results.


**Strength And Weaknesses:**

#Strength
* The paper address a valuable problem: the compounded natural language problems requiring introducing sub-task decomposition and concatenating intermediate supervision
* Sufficient proofs are provided in the appendix

#Weaknesses
* The theoretical analysis of sequence-to-sequence models seem not strong enough, and many of first-mentioned symbols in equations have inadequate explanation. Maybe it is because you have moved many details to appendix. Analogously, the experiments face the same problem.
* The inspiration from this paper is limited, and it seems that the theoretical and experimental results are apparent
* The structure of the paper can be improved. BTW, I think it would be more legible to organize the sequence (1,2,3,4) in a vertical way in Figure 1.


**Summary Of The Paper:**

This paper aims to address the compounded natural language problems required by introducing sub-task decomposition and concatenating intermediate supervision to the input and training the sequence-to-sequence model on this modified input.
The contribution of this paper can be concluded as:
* proving a positive theoretical result on decomposing compounded tasks with neural networks in an end-to-end manner
* The above theoretical results can apply to a broad family of tasks, and currently including SOTA  language models on complex multi-hop tasks in NLP
* generalizing the above results, and demonstrating that sequence-to-sequence tasks with sufficient intermediate supervision  allow learning any function in the P time complexity class



**Summary Of The Review:**

This paper has addressed a valuable problem: sub-task decomposition in sequence-to-sequence learning, and provided theoretical analysis, applied it to tasks and analyzed the time complexity. The paper is easy to follow but the inspiration is imited.

---

> ### Author Response · Authors · 2022-11-18
> **Official Response**
>
> We thank you for your thoughtful feedback and structural comments
>
> 1.
> We thank you for pointing out unclear notations. We have added explanations for symbols that previously appeared in equations without proper explanations in the revised version. We hope that our sequence-to-sequence framework is clearer now.
>
> 2.
> Regarding inspirations from this paper please see our comment for reviewer T2bv.   We emphasize that while our paper does not propose a novel NLP technique, we believe that a solid theoretical foundation for sub-task decomposition is useful as it principally encourages the curation of datasets with intermediate steps for various NLP tasks. Though chain of thought / multihop works are gaining popularity, there is still a prevalent belief that bigger and stronger networks can crack harder and harder tasks. We think that a theoretical result such as ours can bring some principled arguments into the discussion on what can we hope to see solved by larger networks and what needs to be handled differently (by finding ways to insert intermediate supervision).
>
> 3.
> Regarding reproducibility, note that section G in the appendix includes all the experiment details. In addition, we also provided the source code for running the experiment in the supplementary materials.

---

### Official Review · Reviewer_Qt7K · 2022-10-29

**Confidence:** 3
**Correctness:** 4
**Technical Novelty And Significance:** 3
**Empirical Novelty And Significance:** 3
**Recommendation:** 6

**Clarity, Quality, Novelty And Reproducibility:**

Clarity - The paper is well-structured and motivated. The theorems are summarized clearly in the main body of the paper.

Quality - The paper is of sufficiently high quality.

Novelty - The main novelty of this paper is in proving the exponential gap in the learnability of sequence-to-sequence tasks with and without intermediate supervision. The paper builds on Wang et al 2021, Shalev-Schwartz et al. 2017 and Shalev-Schwartz & Shashua 2016.

**Strength And Weaknesses:**

Strengths:

1. The paper proves a useful result shedding light on the effectiveness of breaking down sequence-to-sequence problems into supervised subtasks.
2. The paper is well structured - it studies a well-defined problem, proves theoretical results, and verifies them empirically (albeit in a slightly different setting).

Weaknesses

1. As the authors point out, all of their results hold only when the number of subtasks is O(d) rather than O(1) and in most practical cases, O(1) subtasks are more likely.
2. The theoretical and empirical results use different model architectures.

Questions:

1. Since most of your theoretical claims are for recurrent networks, did you attempt to empirically validate them with RNNs? How do your conclusions hold for RNNs with activations besides ReLU? Is this assumption mostly a result of the setting used in Wang et al. 2021? ReLU RNNs.
2/ Why did you leave the input embeddings and initial hidden states untrained?

**Summary Of The Paper:**

The paper presents theoretical and empirical results supporting an exponential gap between a sequence-to-sequence task and its decomposition into many smaller subtasks. In this study, the authors consider a bit parity task where the objective is to determine if the number of ones in a subset of a 0/1 array of length d is odd or even. This is formulated as a binary classification task.

With this setup, the authors show the following main result:

1) When a task is decomposed into subtasks, a seq2seq model can get a get arbitrarily low zero-one loss (\epsilon) with a number of gradient updates that is polynomial in d.
2) When no supervision over subtasks is provided, for any polynomial time learning algorithm, the zero-one loss will be > ½ - \epsilon.

**Summary Of The Review:**

The main novelty of this paper is in proving the exponential gap in the learnability of sequence-to-sequence tasks with and without intermediate supervision. While there are some limitations on the number of sub-tasks with supervision required, it is still a good starting point for future work that seeks to understand why ideas like “chain-of-though” prompting of language models are so effective.

---

> ### Author Response · Authors · 2022-11-18
> **Official Response**
>
> We thank you for your thoughtful feedback.
>
> 1.
> As we pointed out in the Limitations section of the paper’s Discussion, the amount of sub-tasks in our results is indeed a limitation. That being said, our results still make a significant contribution considering the fact that we are the first to formulate the benefits of sub-task decomposition for learning composite problems with neural networks. Our work motivates a further investigation of the tightness of this assumption.
>
> 2.
> Regarding the fact that the theoretical and empirical results use different model architectures. The aim of the experiments is to reinforce the relevance of our results to common Transformer architectures (Vaswani et al., 2017). Consequently, we did not attempt to run the experiments with RNNs. Having said that, given the size of the observed gap, we believe it should be easy to reproduce our empirical results using RNNs as well.
>
> 3.
> Our assumption of ReLU activations and that the training algorithm leaves the input embeddings and initial hidden states untrained, indeed follows the setting used by Wang et al. These assumptions aim to simplify the analysis of the deviation of the Elman RNN from its infinite width NTK.  They are mainly made for clarity; it is easy to extend their results to the case that the input embeddings and the initial hidden states are trained (as confirmed also by Wang et al).

---

### Decision · Program_Chairs · 2023-01-20

**Decision:**

Accept: poster

**Justification For Why Not Higher Score:**

Subtasks are limited, and the new theoretical results have not led to newly proposed algorithms.

**Justification For Why Not Lower Score:**

There is a clear consensus that this paper should be accepted as a poster, because theoretical results on this topic are rare and novel in the field.

**Metareview: Summary, Strengths And Weaknesses:**

This paper provides some theoretical results on sub-task decomposition for sequence-to-sequence learning problems. All reviewers appreciate the positive theoretical results, which are indeed novel in the context of this line of work. The paper is also well-written and well-organized. Even though there was a question about choosing the fine-tuning vs. in-context learning settings, the area chair still thinks this paper has strong merits to help us understand sequence-to-sequence learning in an earlier setup: Commonly, theoretical results in NLP are behind algorithmic developments. Overall, there was a clear consensus from reviewers that this paper should be accepted.

**Note From Pc:**

if the above contains the word "oral" or "spotlight" please see: "oral" presentation means -> notable-top-5% and "spotlight" means -> notable-top-25%. As stated in our emails, we are disassociating presentation type from AC recommendations

**Summary Of Ac-Reviewer Meeting:**

No AC-reviewer meeting was conducted because it was not classified as a borderline paper.